# A double-pointed wooden throwing stick from Schöningen, Germany: Results and new insights from a multianalytical study

**Annemieke Milks**[1]*, **Jens Lehmann**[2], **Dirk Leder**[2], **Michael Sietz**[3], **Tim Koddenberg**[4], **Utz Böhner**[5], **Volker Wachtendorf**[6], **Thomas Terberger**[2,7]

1 Department of Archaeology, University of Reading, Reading, United Kingdom, 2 Department of Hunter-Gatherer Archaeology, Niedersächsisches Landesamt für Denkmalpflege (Lower Saxony State Office for Cultural Heritage), Hannover, Germany, 3 Archaeological Conservation Department, Niedersächsisches Landesamt für Denkmalpflege (Lower Saxony State Office for Cultural Heritage), Hannover, Germany, 4 Wood Biology and Wood Products, Faculty of Forest Sciences and Forest Ecology, University of Göttingen, Göttingen, Germany, 5 Inventory and Heritage Atlas, Niedersächsisches Landesamt für Denkmalpflege (Lower Saxony State Office for Cultural Heritage), Hannover, Germany, 6 Bundesanstalt für Materialforschung und -prüfung (BAM), Berlin, Germany, 7 Department of Prehistoric Archaeology, University of Göttingen, Göttingen, Germany

* a.g.milks@reading.ac.uk

**Data Availability Statement:** All relevant data are within the paper and its Supporting Information files.

## Abstract

The site of Schöningen (Germany), dated to ca. 300,000 years ago, yielded the earliest large-scale record of humanly-made wooden tools. These include wooden spears and shorter double-pointed sticks, discovered in association with herbivores that were hunted and butchered along a lakeshore. Wooden tools have not been systematically analysed to the same standard as other Palaeolithic technologies, such as lithic or bone tools. Our multianalytical study includes micro-CT scanning, 3-dimensional microscopy, and Fourier transform infrared spectroscopy, supporting a systematic technological and taphonomic analysis, thus setting a new standard for wooden tool analysis. In illustrating the biography of one of Schöningen's double-pointed sticks, we demonstrate new human behaviours for this time period, including sophisticated woodworking techniques. The hominins selected a spruce branch which they then debarked and shaped into an aerodynamic and ergonomic tool. They likely seasoned the wood to avoid cracking and warping. After a long period of use, it was probably lost while hunting, and was then rapidly buried in mud. Taphonomic alterations include damage from trampling, fungal attack, root damage and compression. Through our detailed analysis we show that Middle Pleistocene humans had a rich awareness of raw material properties, and possessed sophisticated woodworking skills. Alongside new detailed morphometrics of the object, an ethnographic review supports a primary function as a throwing stick for hunting, indicating potential hunting strategies and social contexts including for communal hunts involving children. The Schöningen throwing sticks may have been used to strategically disadvantage larger ungulates, potentially from distances of up to 30 metres. They also demonstrate that the hominins were technologically capable of capturing smaller fast prey and avian fauna, a behaviour evidenced at contemporaneous Middle Pleistocene archaeological sites.

**Funding:** T.T. and this project are funded by the Deutsche Forschungsgemeinschaft (DFG, German Research Foundation) – project number 447423357. https://www.dfg.de/ The project is further funded by the Lower Saxony Ministry for Science and Culture, with funds from the Future Lower Saxony Programme of the Volkswagen Foundation – project number ZN3985. https://www.mwk.niedersachsen.de/zukunft.niedersachsen A.M. Is funded by the British Academy Postdoctoral Fellowship PF21/210027. https://www.thebritishacademy.ac.uk/ The funders had no role in the study design, data collection and analysis, decision to publish, or preparation of the manuscript.

**Competing interests:** The authors have declared that no competing interests exist.

## 1. Introduction

Of all the raw materials that Pleistocene humans made use of for material culture, wood is one the most under-represented in the archaeological record. However, discoveries of wooden tools have revolutionised our perspectives on Palaeolithic technologies and hominin behaviours. Noteworthy discoveries include wooden spears and digging sticks that together altered perspectives on Middle Pleistocene hominin diets and hunting [1–3], Late Glacial wooden arrows proving bow-and-arrow hunting during the Palaeolithic [4, 5] and the Late Glacial Shigir idol, which is the earliest monumental sculpture [6]. In spite of the significance of such objects, and perhaps due to their rarity, methods to analyse wooden finds that do survive are underdeveloped, and particularly so in relation to processes of manufacture and use. In this paper, we bring to the forefront new opportunities to analyse and interpret Pleistocene wooden tools through a case study of the first of the wooden tools discovered at Schöningen 13 II-4, a Middle Pleistocene site in Germany dated to ca. 300,000 years ago.

The subject of our study is a relatively short double-pointed wooden stick (ID 1779) which, like most of the other wooden tools from the context at Schöningen known as the 'Spear Horizon', was shaped from spruce (*Picea* sp.) [2, 7]. In contrast to the famous Schöningen spears this complete stick is short, measuring under a metre. Although it was the first significant find, it remained an anomaly until a second stick of similar size and shape was recently discovered in new excavations of the 'Spear Horizon South' [8]. Researchers interpreted both of the shorter double-pointed sticks as 'throwing sticks' (or 'hunting sticks') designed to be thrown rotationally at prey. This functional assessment was based on morphological comparison with ethnographic throwing sticks and supported by the suggestion that surface damage on the newly discovered stick was consistent with impact [2, 8]. Although selective features including growth of the wood, selected working traces, measurements, and morphologies for the more important Schöningen 13 II-4 wooden artefacts are published [2, 7, 8], a detailed macro- and micro-analysis of anatomical features of the raw material, surface traces, fractures, and morphometrics has not been undertaken until now.

Nearly three decades have passed since the initial discoveries of the exceptionally preserved wood assemblage at Schöningen, yet we remain at something of an impasse in understanding the role of wood technologies in Lower and Middle Palaeolithic hominin techno-complexes. Some relevant key sites with wood finds have in the interim period been discovered, and their analytical approaches have advanced our understanding of later Middle Pleistocene Neanderthal wood technologies. In the last five years there has been methodological progress in the application of experimentation, macro- and micro-traceology, and chemical analyses to understand manufacturing and use traces and growth features of prehistoric wood artefacts [e.g. 1, 9–11], although few researchers have addressed taphonomic effects on Pleistocene wood artefacts [but see 12]. Building on this recent work and that of wood specialists working to elaborate on the *chaîne opératoire* of later prehistoric wood [e.g. 13, 14], we present a case study that explores in detail the cultural biography of one of the two double-pointed sticks from Schöningen, inclusive of taphonomic and post-excavation alterations. We do so using multiple analytical techniques that enable detailed surface trace and chemical analyses. These techniques include the application of macro- and microscopic approaches such as 3-dimensional surface microscopy, micro-computed tomography (micro-CT) scans, 3D models, Attenuated total reflection Fourier-transform infrared spectroscopy (ATR-FTIR), and macro-photography. We present the results within a framework that follows the biography of the tool from raw material selection through to excavation and conservation, elaborating on detailed traces attributed to each of its phases over the last ca. 300,000 years.

## 1.1 Background on the site and find

In the autumn of 1994, Hartmut Thieme discovered the Palaeolithic site of Schöningen 13 II in a lignite mine south of Helmstedt (Germany). Subsequent excavations of sediment sequence 4 (Schöningen 13 II-4) yielded faunal remains including butchered animals and flint artefacts [2], while exceptional botanical preservation shows the extent of plant materials available to the Schöningen hominins for technological, subsistence and medicinal purposes [15, 16]. The first discovery of a wooden implement was the double-pointed stick that is the subject of this paper (Fig 1; S1 Fig). During the following years Thieme and the excavation team uncovered further worked wooden objects including the famous Schöningen spears, and interpreted the site as a single mass hunting event [2, 17].

The wooden artefacts from 13 II-4 are located within layers that transition between organic peat and marl sediments (4b-4b/c) [7, 18]. The waterlogged contexts led to the exceptional organic preservation including macro- and micro-fauna, and macro- and micro-botanical remains that together demonstrate that Schöningen 13 II-4 was a lakeshore setting [16, 19, 20]. Tree species previously evidenced in the Spear Horizon include pine (*Pinus*), with dropping levels of birch (*Betula*), and very few alder (*Alnus*), willow (*Salix*), juniper (*Juniperus*) and spruce/larch (*Picea / Larix*) [16, 19–21]. Spruce pollen is sparse in the profile, and is thought to have originated from a significant distance to the lakeshore [19, 20]. Thermoluminescence dating suggests the site formed between ca. 337–300 ka BP corresponding to the end of interglacial Marine Isotope Stage (MIS) 9, in a context of deteriorating climatic conditions [22]. Butchered animal remains are primarily from horse (*Equus mosbachensis*), but also include species such as red deer (*Cervus elaphus*), and bovids [23–25]. The Schöningen hominins also used expedient bone tools, including use of the humerus of a saber-toothed cat (*Homotherium latidens*) as a lithic retoucher [26, 27]. Lithic tools at the site are primarily assigned to a Lower Palaeolithic technology and include retouched and unretouched flakes from frost-shattered

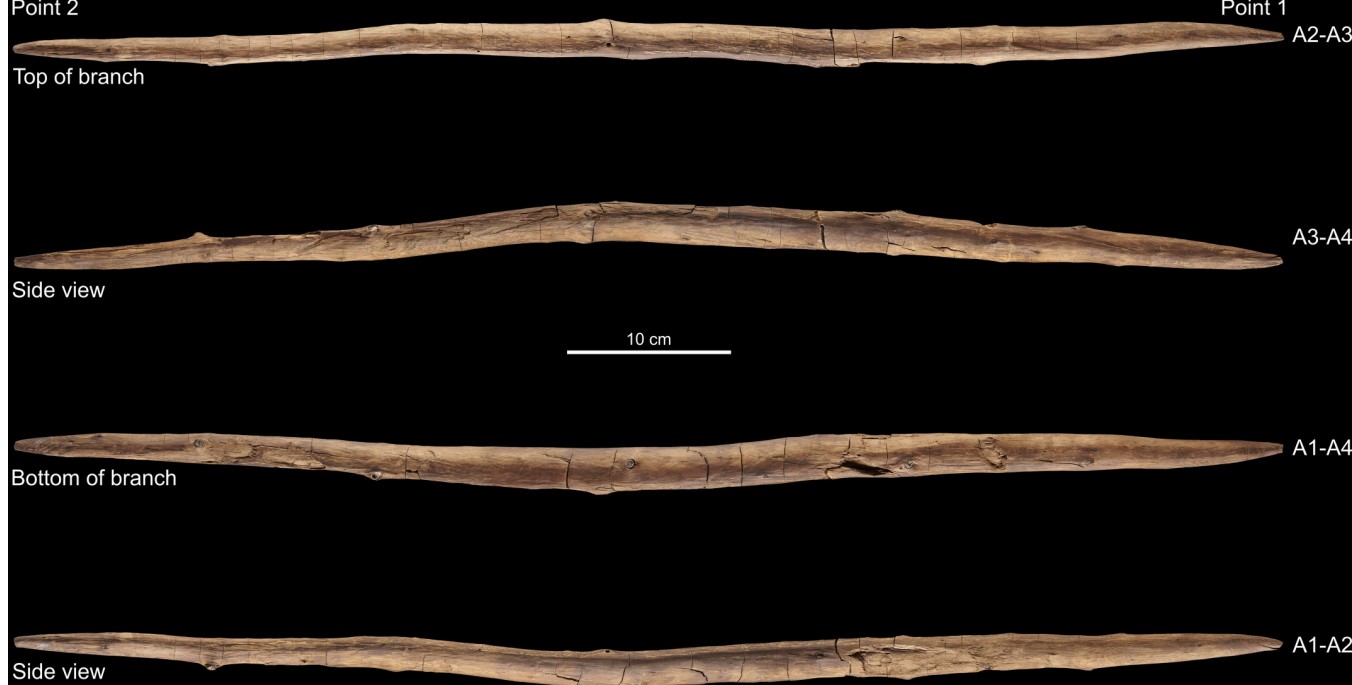

**Fig 1. Overview photo of the double-pointed stick.** Photos: V. Minkus.

flint blanks, some of which show use-wear consistent with woodworking taking place at the site [28–30]. Subsequent sedimentary pressure and faulting resulted in some deformation of nearly all the wooden finds, which were likely susceptible to alteration due to their waterlogged condition [7]. Evidence that Eurasian Middle Pleistocene hominins controlled fire comes from hearths at sites dated to MIS 13 and MIS 11 [31–34]. While it was initially thought that Schöningen had evidence of both hearths and burnt wooden implements, subsequent analyses provided support for neither [35]. On the other hand, in that study only a single piece of wood from an earlier site (Schöningen 12 B) was subjected to analysis, and the question of use of fire to shape and/or alter the wood at Schöningen remains an open one.

Today the wooden artefacts from Schöningen 13 II-4 consist of complete and fragmented tools including at least ten spears, manufactured from spruce (n = 9; *Picea* sp.) and pine (n = 1; *Pinus sylvestris*) [7]. The complete spears are double-pointed, and published data report them as being between ca. 184 cm and ca. 253 cm in length and between ca. 2.3 cm and ca. 4.7 cm in maximum diameter [7]. Spear II has been experimentally tested, with results demonstrating its functionality as both a thrusting and hand-thrown spear [36–39]. Spear VI is shaped differently to the other spears, as whilst it is also double-pointed it has a significant natural kink, making it unlikely to have functioned as a projectile [7]. In addition to the larger spears there are two shorter double-pointed sticks shaped from spruce, and hundreds of smaller wet wood fragments that are currently undergoing analysis. The double-pointed stick analysed in this paper (ID 1779) is now in two fragments but was originally fitted together *in situ* (Fig 2). It was the only wooden artefact in square meter x 684/ y 31, is located 16 metres away from the nearest spears (Fig 3) and ca. 120 m from the second double-pointed stick (square meter x 772/ y -49) [8, 18]. After discovery, the piece was conserved using Kauramin 800™, a mixture of oligomeric and monomeric forms of melamine and formaldehyde [40], and is currently on display at the Forschungsmuseum Schöningen (Schöningen, Germany). No permits were required for the described study, which complied with all relevant regulations. Both of the short double-pointed tools, measuring under 1 metre, have various functional interpretations, but are most often viewed as 'throwing sticks' [2, 8, 41–43]. The artefact analysed here is hypothesised to have potential additional or alternative functions including as a short thrusting or stabbing weapon [2], a digging stick [7], a bark peeler [44], or a child's spear [7].

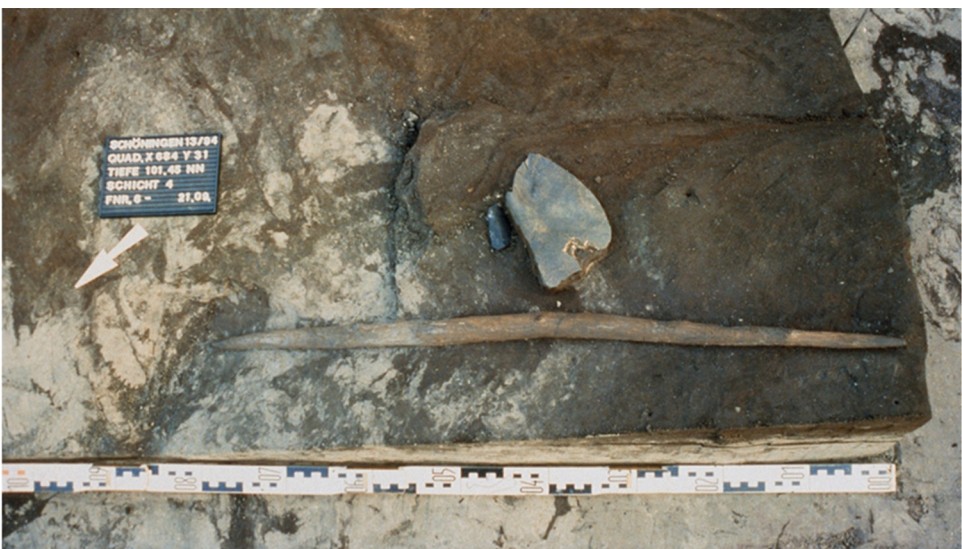

**Fig 2. Excavation photograph of the double-pointed stick.** Photo by Peter Pfarr.

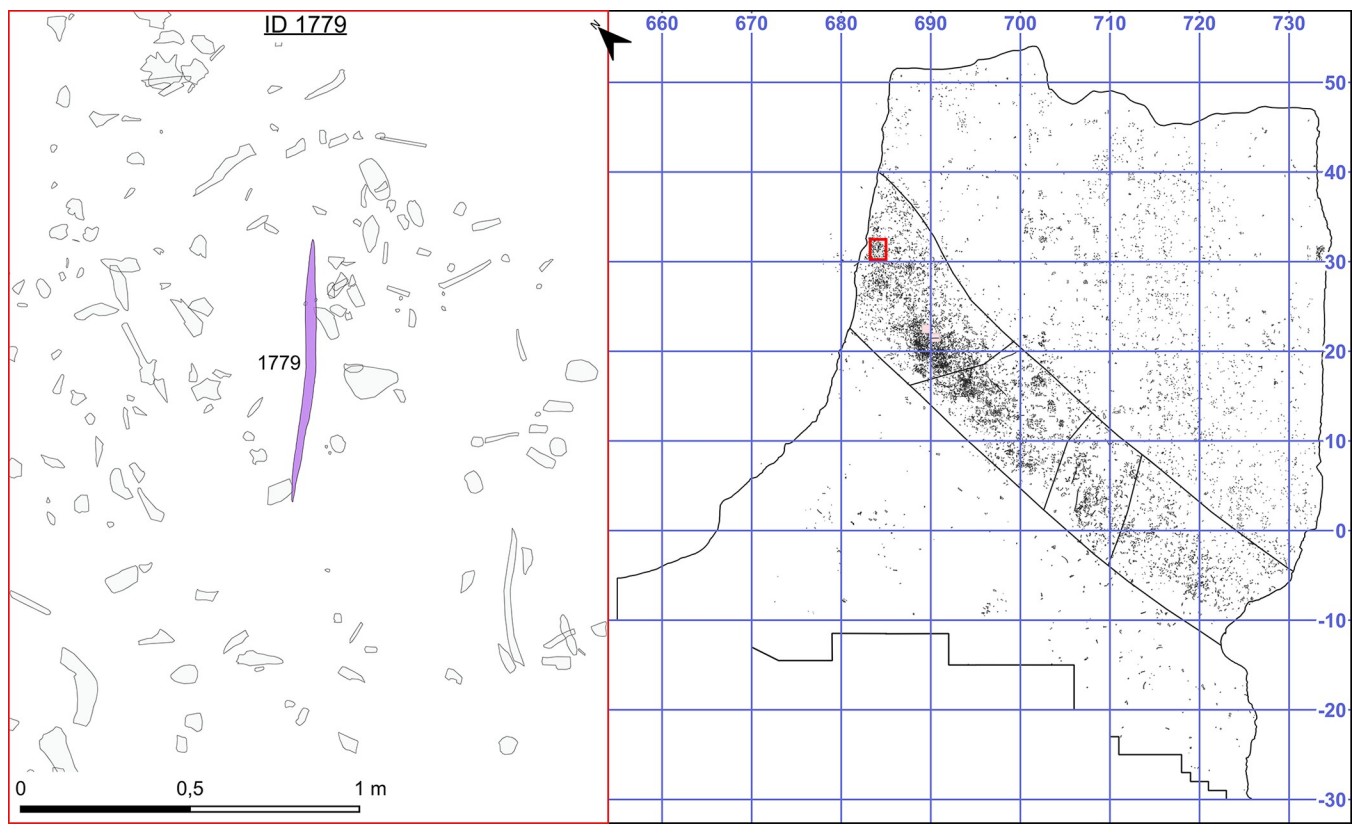

**Fig 3. Location of the find in the wider site context.** Image by D. Leder.

## 2. Materials and methods

### 2.1. Structural, morphometric and trace analysis

The non-destructive analyses of traces comprised a combination of methods developed on the basis of existing approaches to wood and bone tools, including macro- and micro-analyses. The length measurement was taken as a straight line between two points in View A1. Sedimentary pressure altered the cross-sectional shape along most of the length of the tool. To account for this, diameter measurements were taken using callipers in two directions (A1–A3 and A2–A4) and are reported as averages, accounting for differences from previously reported metrics of the object. Morphological descriptions, further details of which are found in S1 File, follow Bordes [43]. The artefact was investigated on a macro-scale using variable light sources with the naked eye and a loupe. Macro photographs (labelled in the manuscript as taken by Volker Minkus) were taken using a Nikon D850 with 50mm Sigma-ART and Nikon 105mm lenses (resolution of 45.7 megapixels). Markers located beside the artefact helped to find the correct positioning. Subsequent development of the images was undertaken using Helicon Focus, Capture One and Adobe Photoshop software.

A high-resolution 3D model was created by GOM Metrology to examine and present surface features and overall tool morphology. Micro-traces were investigated and documented with the use of a stereo microscope (LEICA M80). Micro-computed tomography (micro-CT) scanning was performed by Waygate Technologies using a phoenix Vtomexm enabling the investigation of internal growth and taphonomic features, as well as elucidating the manufacturing process. Topographic surface imaging of selected surface features was

accomplished using the 3D reflected light microscopy Keyence VHX-5000. Based on such topographic 3D images, morphometric analyses (e.g. profile lines) of selected surface features were also performed enabling further evaluation of the causes of features such as striations. The volume of the two fragments was calculated from the 3D model using Artec Studio 16 3D Software and Geomagic® Essentials. Detailed information about the methods used for the 3D surface analyses, micro-CT scanning and 3D model is found in S1 File, sections 1.1, 1.2 and 1.4 respectively. All traces were mapped onto a digital drawing of the double-pointed stick.

## 2.2 Residue analysis

Apart from structural imaging, chemical analysis of a selected dark patch was performed using non-destructive Attenuated total reflection Fourier-transform infrared spectroscopy (ATR-F-TIR) to understand whether these areas are residues that could have been formed by mould or by deliberate charring. FTIR was selected based on previous applications of this wood and wood components [45–49] and charring [50–52]. We selected View A2 on Point 1, concentrating on seven darker areas near the point, and three visually lighter coloured areas near the centre of the artefact. Detailed information about the materials and methods for the residue analysis is in S1 File, section 1.3.

## 2.3 Wood species identification

The identification of wood species follows established botanical methods. Tree ring sequences and analyses were reconstructed with the use of micro-cuts and micro-CT scans. The micro-cuts were documented with a digital microscope (LEICA M125c reflected light microscope, planapochromatic objectives 0.63x and 1.6x), MC170 HD CAMERA, CLS 150 LED ring light and LEICA-LAS software V4. 12.0. These approaches enable an estimation of the minimum age of the branch and thus tree, the identification of growth characteristics, and other features such as compression wood. In addition, natural material and standard reference texts [e.g. 53] were consulted for interpreting natural features.

## 2.4 Establishment of views

The object was analysed on four planes, labelled Views A1 through A4. View 1 is known as A1 (for German designation 'Ansicht 1') and is defined for the larger Schöningen wood objects by the surface that was uppermost in excavation. Because it is a 3D object in the round and the views as originally defined do not correspond to the way the wood grew, we often refer to the surface between two views (e.g. View A1–A2). The two extremities are designated as Point 1 and Point 2 (Fig 1). In order to remain consistent with previous research and reports, this study follows the original measurement system, with 0 cm starting from Point 2.

## 2.5 *Chaîne opératoire* approach and nomenclature

This paper uses a new standardised reference glossary with defined terms organised within a *chaîne opératoire* framework, which we have recently published [54]. The reader is referred to this glossary for definitions of terms used throughout the paper (e.g. branch whorl, tangential annual rings, tool facet, mineralisation). The cultural biography of the artefact is explored through five phases from raw material sourcing (Phase 0), manufacture (Phase 1), use, maintenance and discard (Phase 2), taphonomy (Phase 3), and excavation and post-excavation (Phase 4) (see Table S3 in S1 File for a more detailed description of each phase). Methods to evaluate potential use traces (Phase 2) on Pleistocene wood tools, and to reliably distinguish them from manufacturing and taphonomic damage are underdeveloped [but see 55]. To

facilitate analysis of use traces and fractures we used an experimental reference collection of spruce spear replicas used for thrusting and throwing [36, 37, 39]. To assess taphonomic factors we used existing publications on natural wet wood [12, 56] alongside new observations.

### 2.6 Ethnographic analogy

To better understand characteristics and contexts of 'hunting sticks' (also called 'throwing sticks') we conducted a systematic review of ethnographic literature using keywords to search in the eHRAF database. Results were coded for location, use, prey type and morphometrics (see S1 File, section 1.6 for detailed methods). The full results and sources are available as an accompanying dataset (S4 Dataset).

## 3. Results

### 3.1 Phase 0: Raw material

The double-pointed stick was manufactured using spruce (*Picea* sp.) (see also 2). The use of spruce is in keeping with the wider sample of worked wood from the find horizon (13-II 4b and 4c). Ongoing species idenfication the wood material by one of the authors (M.S.) as part of the current research project demonstrates that in this same find horizon hominins also exploited pine (*Pinus sylvestris*) and larch (*Larix*), alongside a background presence of willow (*Salix*) and/or poplar (*Populus*), species which do not show signs of human modification. According to palynological analyses, spruce did not belong to the natural background vegetation at the site and the raw material was introduced by hominins [16, 19, 20].

The blank (i.e. the piece of wood prior to preparation and shaping) [e.g. 57] selected for the tool was a branch. We determined this along several lines of evidence like arrangement of branches, absence of branch seams and whorls, and presence of compression wood. The double-pointed stick exhibits a natural curvature in the direction of View A1–A4 curving towards View A2–A3 (Fig 1). Morphologically, spruce branches typically have a natural upward curvature (Fig S5 in S1 File). We counted 47 knots encompassing both previously living and dead limbs along the length of the tool, and these are asymmetrically placed. The vast majority of knots are located on View A2–A3, which can also be explained by the deeper working of the wood and the associated cutting of older, already overgrown knots, followed by a smaller number of knots on views A1–A2 and A3–A4, alongside a dearth of branches on view A1–A4 (Table 1). The asymmetrical placement of knots along the view A2–A3 corresponding to the curvature suggests that this was the top of a naturally curving branch. The absence of seams and of clear, organised and symmetrical branch whorls argues against it being a trunk. Branch whorls are an arrangement of three or more branches at the same height around the trunk of conifer trees (Fig S4 in S1 File), and they appear star-shaped in cross-section [54]. In contrast, branch shoots emerging from parent branches have a different growth pattern, and tend to be stronger and more numerous along the parent branch's top and sides due to exposure to light. Second, the wood displays a curvature which is most visible from View A3–A4 (Fig 1).

**Table 1. Location of knots on the four views.**

| View | Knots |
| --- | --- |
| A1–A2 | 10 |
| A2–A3 | 18 |
| A3–A4 | 10 |
| A4–A1 | 9 |

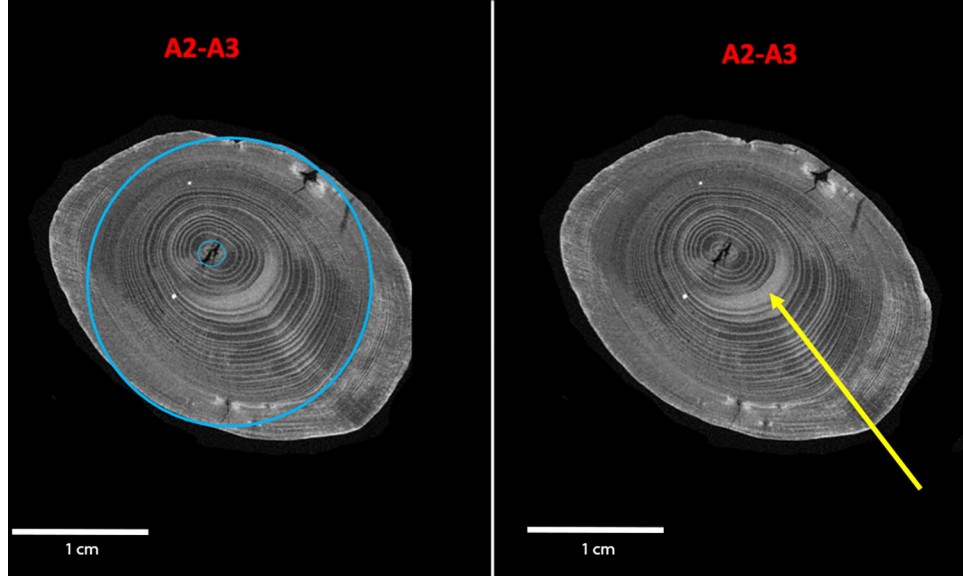

**Fig 4. Micro-CT scan slice of an area without surface damage (45 cm).** The blue circles in the left image show that the central pith, the smaller blue circle, is eccentric, i.e. not centrally placed in the cross-section. The arrow in the right image points to compression wood (the term for reaction wood in coniferous species) with wider annual rings on the underside of the branch. micro-CT scans: Waygate Technologies. Image: A. Milks.

Additional growth characteristics are evidenced through the tree ring analyses. Tree rings are wider on one side of the blank in relation to the central pith, and the growth may also be eccentric, with the pith outside of the central axis of the cross-section (Fig 4). In the wood of stems and branches, eccentric growth can be caused by the formation of reaction wood. For gymnosperms eccentricity can occur as a result of the development of compression wood on the underside of branches. The effects of gravity are counteracted by the branch by developing denser and harder wood on the underside, often resulting in an area with wider latewood in that area [53, 54, 58, 59]. Thereby, the wide and dense latewood regions in the compression wood appear in distinct bright grayscale intensities in micro-CT images (Fig 4) with more structural uniformity compared with the normal/opposite wood [60]. On the double-pointed stick, the eccentricity of the pith is indicated in the cross-sections along the entire length of the piece. This eccentricity is illustrated in a micro-CT slice from the midsection of the tool, where it was only minimally debarked and otherwise undamaged, and thus the off-centre location is not a result of working away more material on selected faces. The double-pointed stick exhibits a natural curvature in the direction of View A1–A4 curving towards View A2–A3. The characteristics, including an absence of whorls, distribution of knots, presence and location of compression wood in comparison with the curvature, pith eccentricity, and natural curvature demonstrate that the blank selected was a branch.

A total of 63 tree rings were counted on the artefact at 47 cm, representing a very slow growth rate of ca. 0.2 mm per annum (see also S1 Dataset). In the centre of the wood the year ring width is subject to strong fluctuations. Annual rings 8, 15, and 17 for example are very narrow, indicating deteriorating growing conditions in the summer months, with little latewood formation (Fig 5B and 5C). In the region of tree ring 10 (Fig 5C, arrow 3), there is compaction of the cell structure, which can be interpreted as the premature latewood formation of the branch during that summer. After a temporary deterioration in growth conditions, possibly due to severe drought, the branch continued to grow again. After a rapid decrease in the

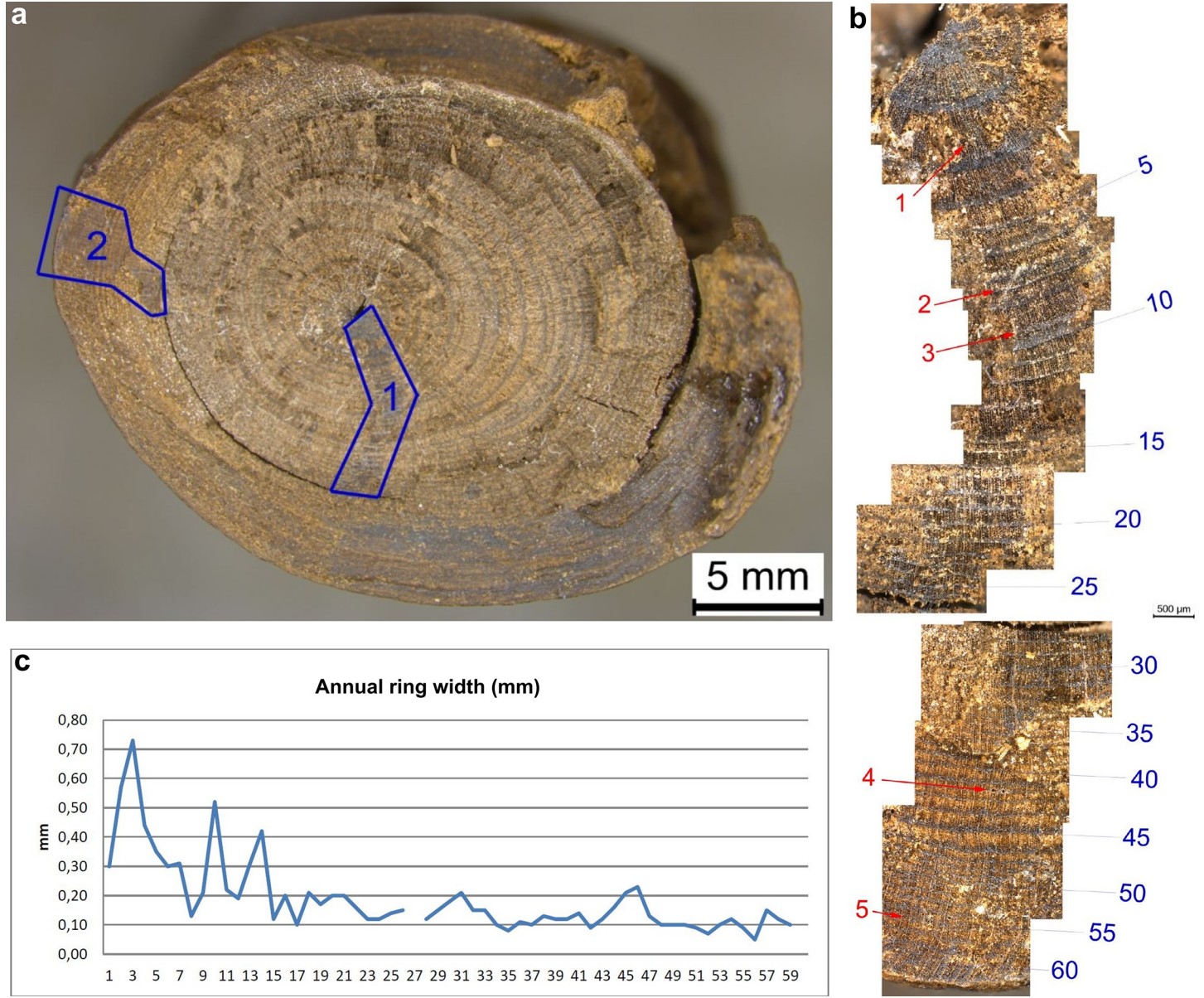

**Fig 5. Annual ring analysis.** a. Taphonomic transverse break surface, from where micro-cuts were taken. b. Composite image of different selections of the cross-section micro-cuts (corresponding to Area 1 on the left image). Blue numbers indicate the annual ring. Red arrows highlight features mentioned in the text. c. graph showing measurements (mm) of each annual ring. Images M. Sietz.

width of the tree rings within the first 15 years of growth of the branch, they rarely exceed the 0.2 mm mark (Fig 5C). An extremely small growth in thickness in years 35, 42, 51, 52 and 56 of less than 0.1 mm indicates poor growth conditions, and the narrowest of these, annual ring 56, has a width of only 0.05 mm, consisting of only 2 or 3 cells (Fig 5B, red arrow 5).

In conifers, small tree rings result in higher-density wood [53]. The narrow growth rings on the artefact reflect poor growth conditions which could result from a response to climate for example, growth at higher altitudes and/or sites with shortened growing seasons, or nutrient deficiencies [61]. Alternative causes for the continuous decrease in the outer annual ring widths seen on the artefact could be due to it being an older branch (Fig 6) [62], climatic deterioration,

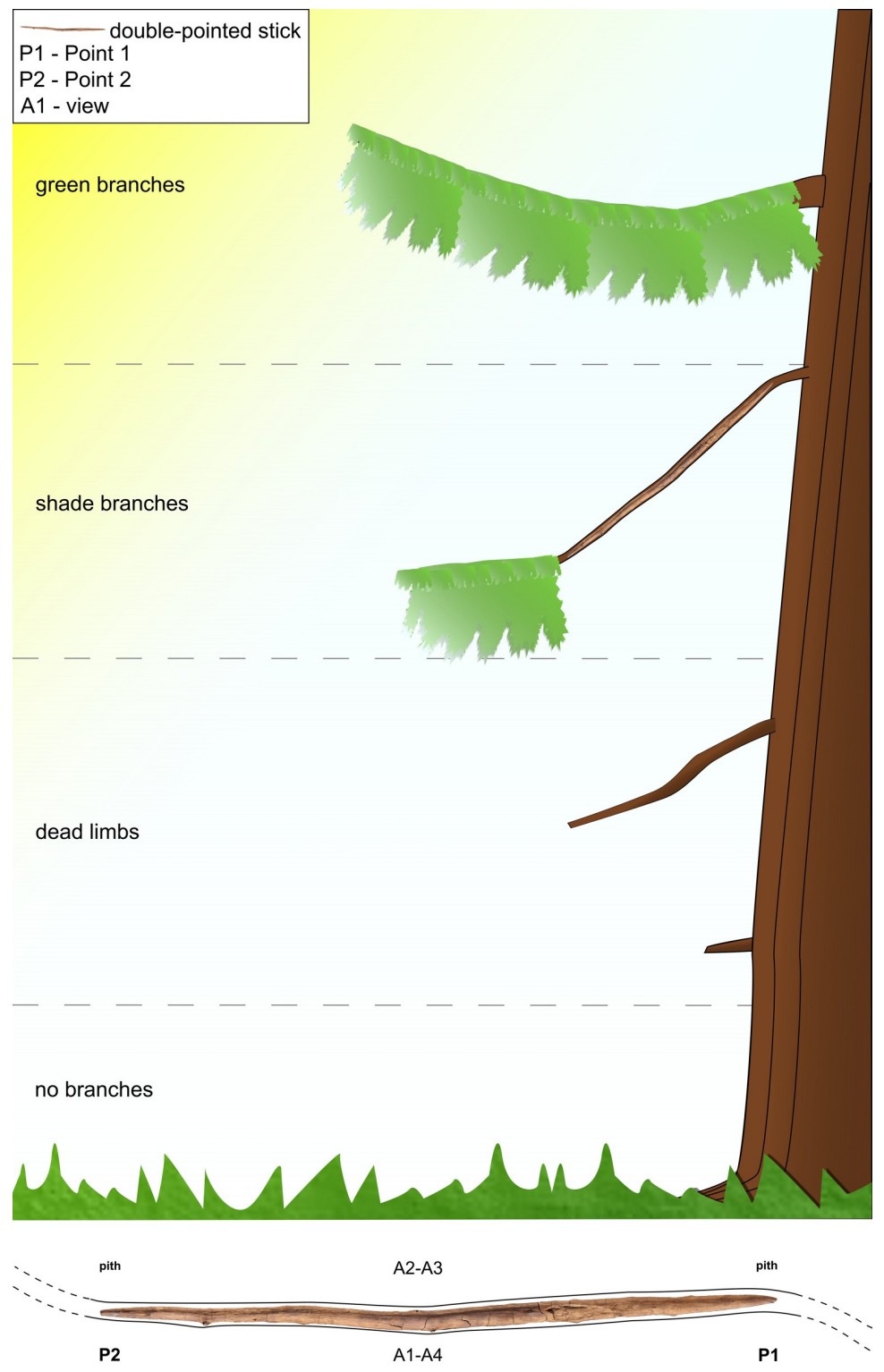

**Fig 6. Schematic reconstruction of the double-pointed stick in relation to the raw material selection.** Image by D. Leder.

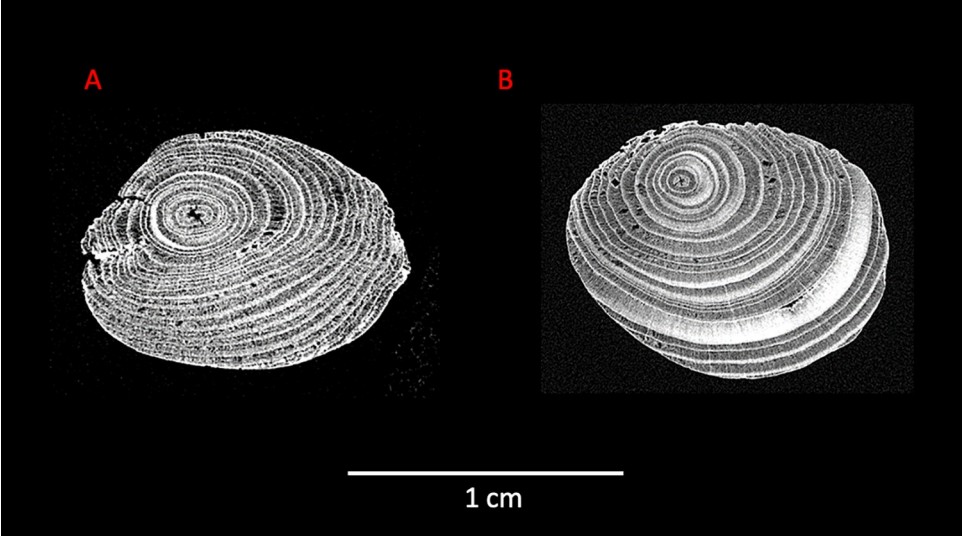

**Fig 7. Micro-CT scans of Points 1 and 2.** A: micro-CT scan of Point 2 (2 cm). B: micro-CT scan of Point 1 (75 cm). micro-CT scans: Waygate Technologies.

infestation of the tree by fungi or insects, or a natural process of the branch dying off. Comparing the growth ring distances in different locations shows a slight narrowing towards Point 2, suggesting that Point 1 was where the branch attached to the trunk (Fig 7). The interpretation of where the branch attached to the trunk is further supported by the fact that dead knots and knot holes are located between 22 cm and 62 cm (Table S4 in S1 File), with 75% located in the Point 1 half of the stick. This indicates that the direction of growth was from Point 1 towards Point 2, with older branch shoots and/or those subjected to more shade most likely to die off. Finally, the initial direction of the growth of auxiliary branches in the direction of Point 2 can be seen in a longitudinal slice from the micro-CT scan (Fig S6 in S1 File).

## 3.2 Phase 1: Manufacture

**3.2.1 Morphometric description.** Today the stick measures 77.2 cm in length, with a maximum diameter of 2.5 cm (for detailed measurements see S2 Dataset). The location of the maximum diameter is at 47.2 cm, or 61% of the length measured from 0 cm (Point 2). The total volume of the two fragments together equals 239 cm$^3$. Depending on the growing conditions such as climate, modern Norway spruce (*Picea abies*) can have air-dry density values ranging from 0.232 g/cm$^3$ to 0.588 g/cm$^3$ and a mean value of 0.344 g/cm$^3$ (n = 368) [63]. Spruce growing in today's modern temperate conditions tend to have wider annual rings and lower density than the double-pointed stick. For example, a sample with a mean annual ring width of 1.44 mm corresponded to an air-dry density of 0.460 g/cm$^3$ [64]. With annual rings considerably narrower on the double-pointed stick (mean = 0.2 mm), this would likely result in it having an higher density compared with modern spruce [65]. On the basis of the upper limit of modern spruce density and considering the determined volume, we estimate the original mass of the tool to have been ca. 141 g.

Morphologically both points are shaped similarly and are of similar dimensions. However, although the two ends of both points are of similar size and shape, the lengths of the two tapers differ. The taper towards Point 2 is longer (ca. 45 cm) than that of Point 1 (ca. 22 cm) (Fig 8). There is a very slight natural taper from Point 1 towards Point 2. This natural taper was

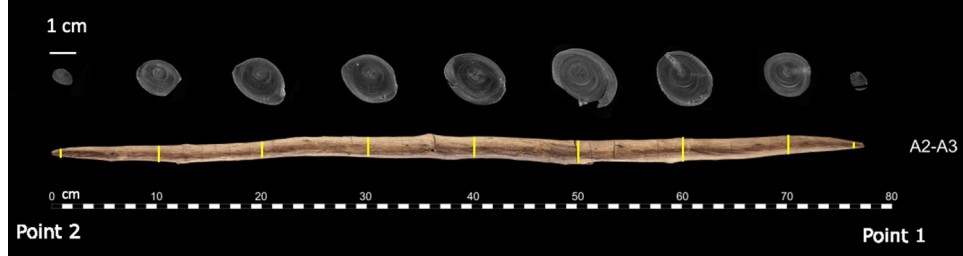

**Fig 8. Composite image of the micro-CT slices at regular intervals, illustrating the change in diameter and shaping strategy.** Photo: V. Minkus. micro-CT scans: Waygate Technologies and T. Koddenberg. Illustration: A. Milks.

measured by calculating the distance between the pith and annual ring number 40 at two locations (57.2 cm and 20 cm). The difference in distance between these locations is only 2.27 mm. In contrast, the difference in diameter between the maximum diameter (47 cm) and 20 cm is 7.75 mm. Furthermore, while most of the shaft of Point 1 consists of a debarked surface, along Point 2 (ca. 33.5 cm towards 0 cm) we see clear tangential annual rings on multiple surfaces (see also 3.2.2; Figs 9 and 10) reflecting deeper working into the wood in order to taper the shaft. In contrast, Point 1 only has annual ring surfaces measuring less than half the length of those of Point 2 (12.7 cm long, from 64.5 cm to 77.2 cm on Views A1–A4 and A1–A2). Therefore, material was deliberately removed from Point 2 over a longer section of the shaft than from Point 1, intentionally creating an asymmetrically shaped tool.

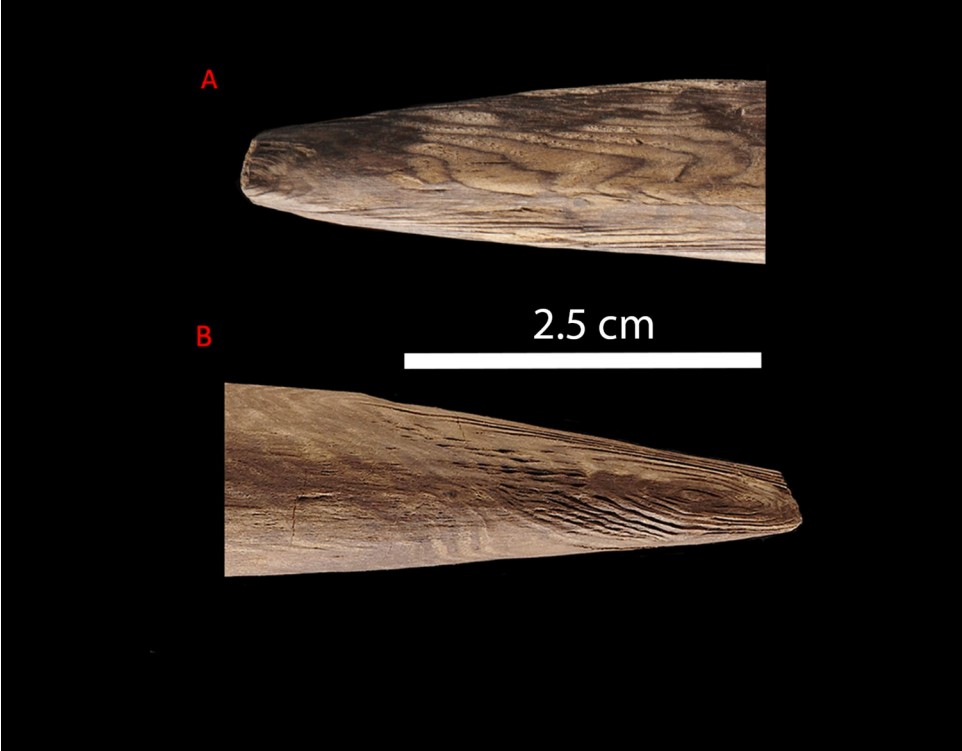

**Fig 9. Detail of Points 1 and 2.** A: detail of Point 2 (A2–A3), on the top side of the branch. Note that the tip is broken off just before the pith emerges at the surface. B: detail of Point 1 (A2–A3), with clearly visible annual rings visible due to working, the bevelled shape clearly evidenced, and the pith emerging on the top of the branch. Photos by V. Minkus.

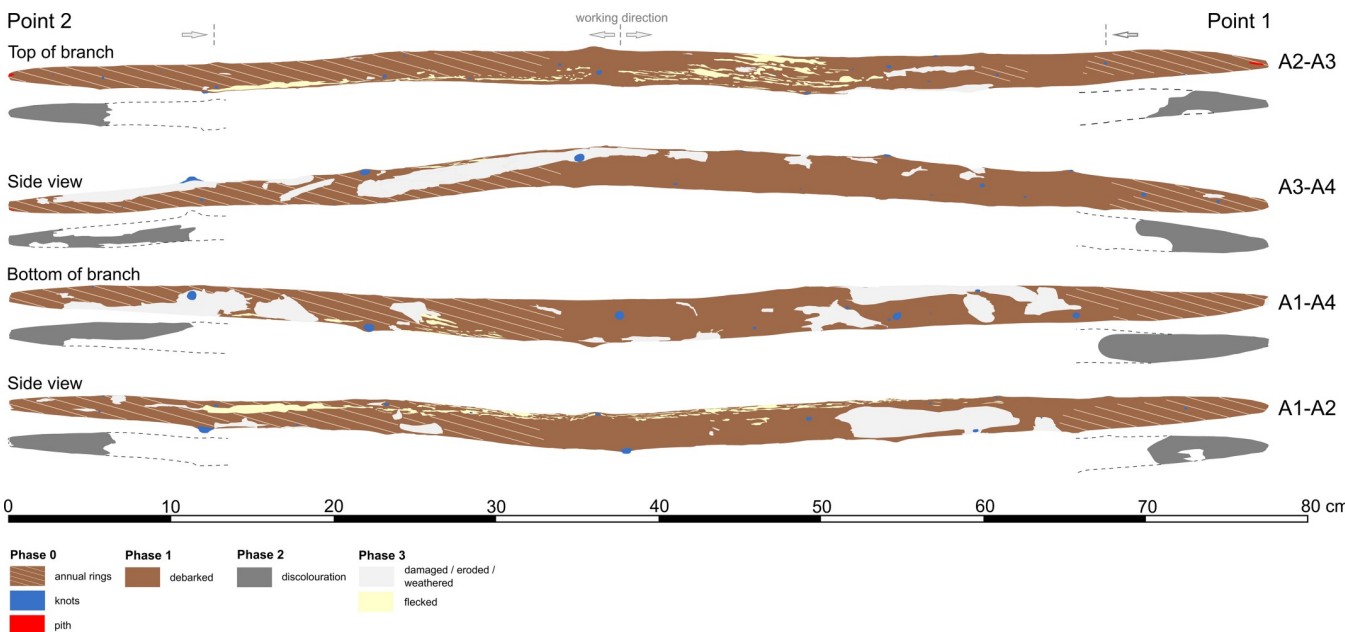

**Fig 10. Schematic simplified mapping all of the relevant traces from phase 1 through phase 4.** See Fig S19 in S1 File for a more detailed mapping. Drawing D. Leder.

To shape the ends of each point the toolmaker did not follow the natural curvature of the branch, but rather worked away more wood from the top of the branch. The result is that like the larger spears, the softer pith emerges along the side (View A2–A3) rather than directly at the tips (Fig 7). This is visible in the tangential and radial ring areas on both points (Fig 9) and in the tangential cross-sections (Fig 7). The finished tool is slightly curved (visible in Fig 1, side views A3–A4, A1–A2; see also S1 Fig). While the current cross-sectional profile of the shaft is slightly elliptical (Fig 8) this is primarily due to natural growth, although there are indications that taphonomic compression also played a minor role (see also 3.4). In contrast, both points are clearly intentionally shaped to create ends with a slightly more elliptical cross-section (Fig 7).

**3.2.2 Manufacturing traces.** The tool is fully debarked, with no outer bark, inner bark or cambium (Fig 10). A series of striations oriented obliquely to the shaft have profiles and lengths (ca. 5 mm to 15 mm) consistent with cut marks (Fig 11; Table S5 in S1 File). These cut marks likely facilitated the debarking process, allowing the toolmaker to cut into the bark and pull it off in strips when relatively fresh. The longest cut mark is arc-shaped and is over 50 mm long (View A2–A3 from 14.5 cm to 19.5 cm; Fig S7 in S1 File). Micro-analyses show this mark has fibre deformation and a profile consistent with an angled cut with a sharp tool edge (Fig 12). In general the 3D microscopy demonstrates morphometric variability (Table S9 in S1 File; S3 Dataset). Organised groups of longitudinal parallel and sub-parallel striations are likely scraping marks (Fig 13) to remove any remaining bark tissue, and to regularise the surface of the tool. In contrast to the points where the wood is worked more deeply, much of the shaft closely follows the course of the debarked surface of the outer annual ring, with these areas characterised by the presence of scraping marks and abraded areas. A small number of tool facets with stop marks and signatures represent deeper carving or possibly planing (e.g. Fig 14; Table S5 in S1 File). The stop marks show the working direction; in the centre of the shaft the working direction was towards the points, while at both points the working direction was towards the centre (Fig 10).

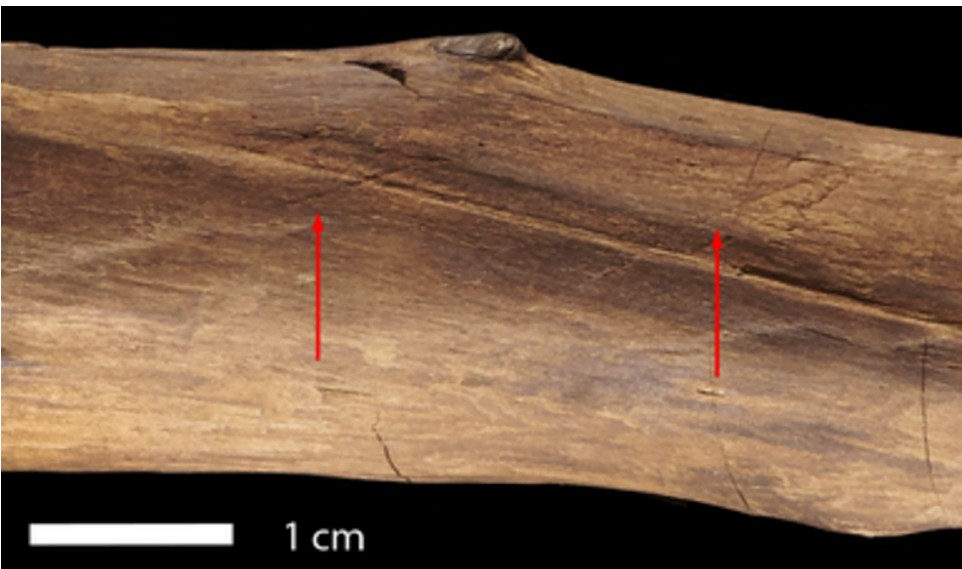

**Fig 11. Examples of short, obliquely oriented cut marks.** View A3–A4 at 62.5–69 cm. Photo V. Minkus.

The toolmaker created the tapers of both points by carving or planing, in the process creating very long surface facets, which measure over ca. 35 cm on Point 2 (View A2–A3) and ca. 10–12 cm on Point 1 (View A1–A4) (Fig S8B and S8D in S1 File). Surface facet edges are rounded, and were likely smoothed by abrasion, and are visually distinct from edges created through taphonomic compression. We detected few clear tool marks in the annual ring areas, attributed to rough carving and/or planing. Finer tool marks are visible on other areas and with a general absence of taphonomic smoothing (see Section 3.4), these areas are interpreted as being deliberately abraded. A particular focus was paid to working knots (Fig 15). Knots are particularly hard to work, and the toolmaker shaped all but two of them down to be flush with the surface (Fig 15A, 15B, 15D–15G; Fig S19 in S1 File). The aim of this was likely to improve

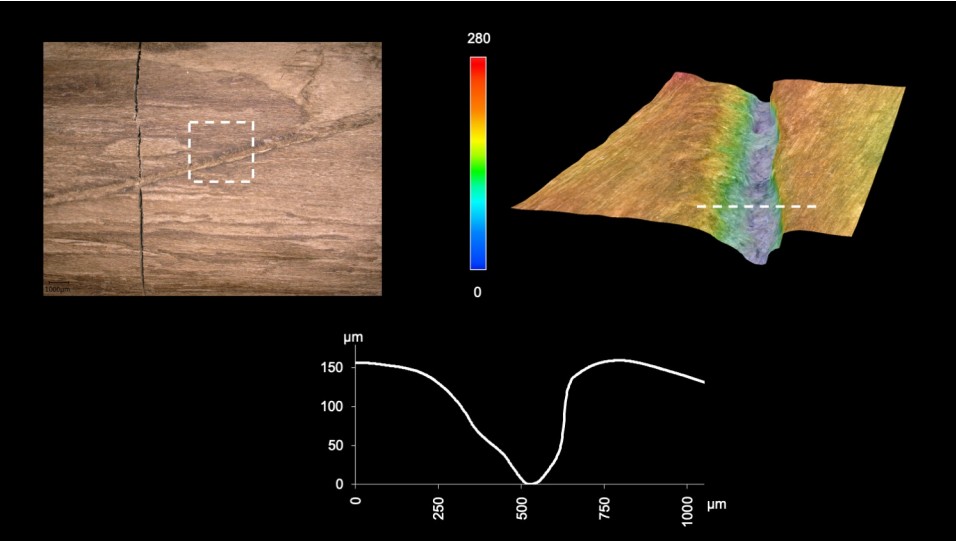

**Fig 12. 3D microscopy of an example cut mark.** View A3 at 16.5–18 cm. Images T. Koddenberg.

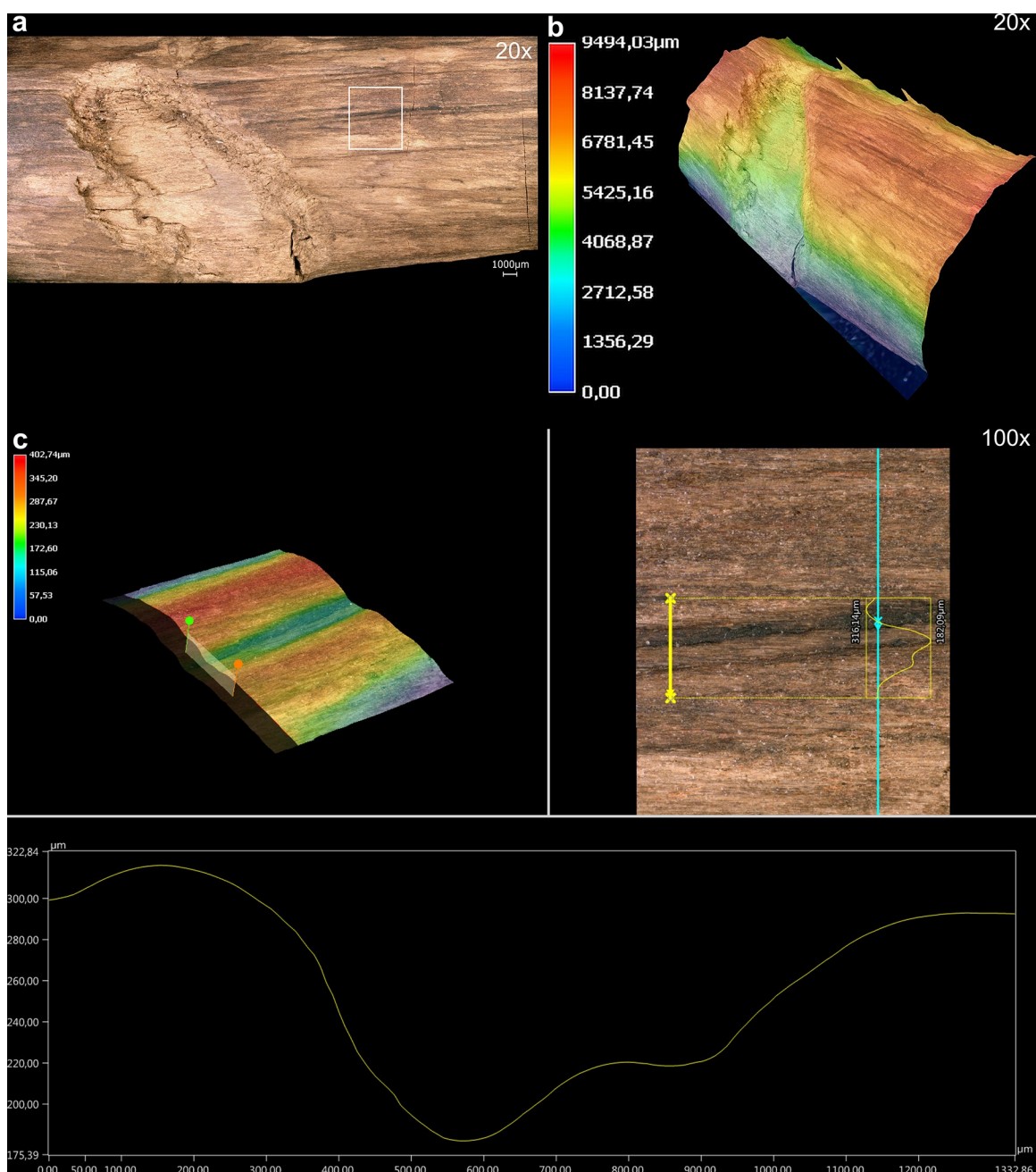

**Fig 13. 3D microscopy of an example of an area with long, shallow scraping marks.** View A1 at 57–61 cm. Images T. Koddenberg.

handling for ergonomic purposes, and to improve aerodynamics by reducing drag. An absence of significant surface or internal drying cracks suggests the wood dried slowly and evenly. Cut wood loses its natural moisture until it is in equilibrium with the surrounding environment, and if freshly cut and debarked wood is allowed to dry too quickly it can develop significant cracks and can also warp [53]. Seasoning to remove moisture at a uniform rate can be accomplished in different ways, including allowing a branch or trunk to dry slowly with the bark attached, storing the wood in such a way as to allow uniform air circulation, storing it with worked wood chips, or over a fire to dry the wood.

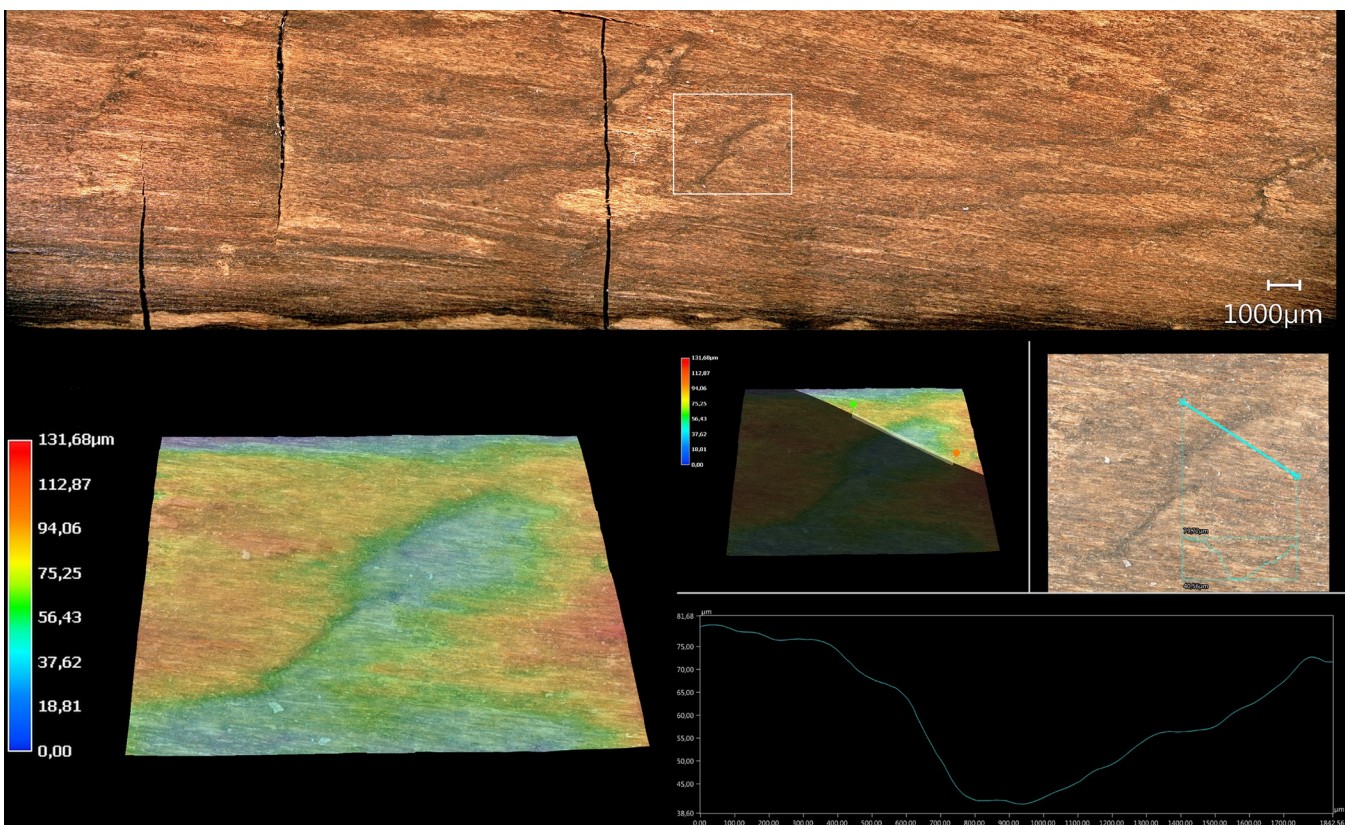

**Fig 14. 3D microscopy of an example of a tool facet with stop mark.** View A3, 45–46 cm. Images T. Koddenberg.

One of the characteristic features of the double-pointed stick is black staining around the perimeter of both points (Fig 10) which was more visually evident prior to conservation treatments (Fig S9 in S1 File). This might be connected to Phase 1 including the potential application of a material such as grease or wax to seal the wood, use of fire to shape the points, or heat treatment of the points over an open fire, with the latter two processes leaving behind soot and charring residues. These traces could also have been left during use (Phase 2), for example from contact with animal fat or blood. Taphonomic factors (Phase 3) can be excluded because in spite of the artefact being embedded fully within the same sedimentary context, the discoloration is restricted to the points. Furthermore, the darker areas on both points also do not faithfully follow the annual ring areas, meaning that this is not from differential staining of exposed heartwood.

The ATR-FTIR measurements of the double-pointed stick 1-A2 showed wide variation of the spectra obtained and therefore all measurements of light and dark spots respectively were averaged (Figs S10 and S11 in S1 File). All spectra of the throwing stick showed being influenced by the embedding agent, Kauramin 800 (differentiating absorption at 909 cm$^{-1}$). The results showed that IR-spectra of light and dark measurement spots are very similar. The very few differences in absorptions between dark and light areas, (e.g. 1718 cm$^{-1}$, 1644 cm$^{-1}$, and 1213 cm$^{-1}$), are within the width of standard deviation of measurement points and hence are not significant. It was not possible to detect changes on the wood surfaces that resulted from exposure in fire. This is likely due to a number of confounding factors including that wood spectra are a mixture of the main components lignin and (hemi-) cellulose, and growth rings result in varying proportions of these components and in a resulting modulation of the spectra.

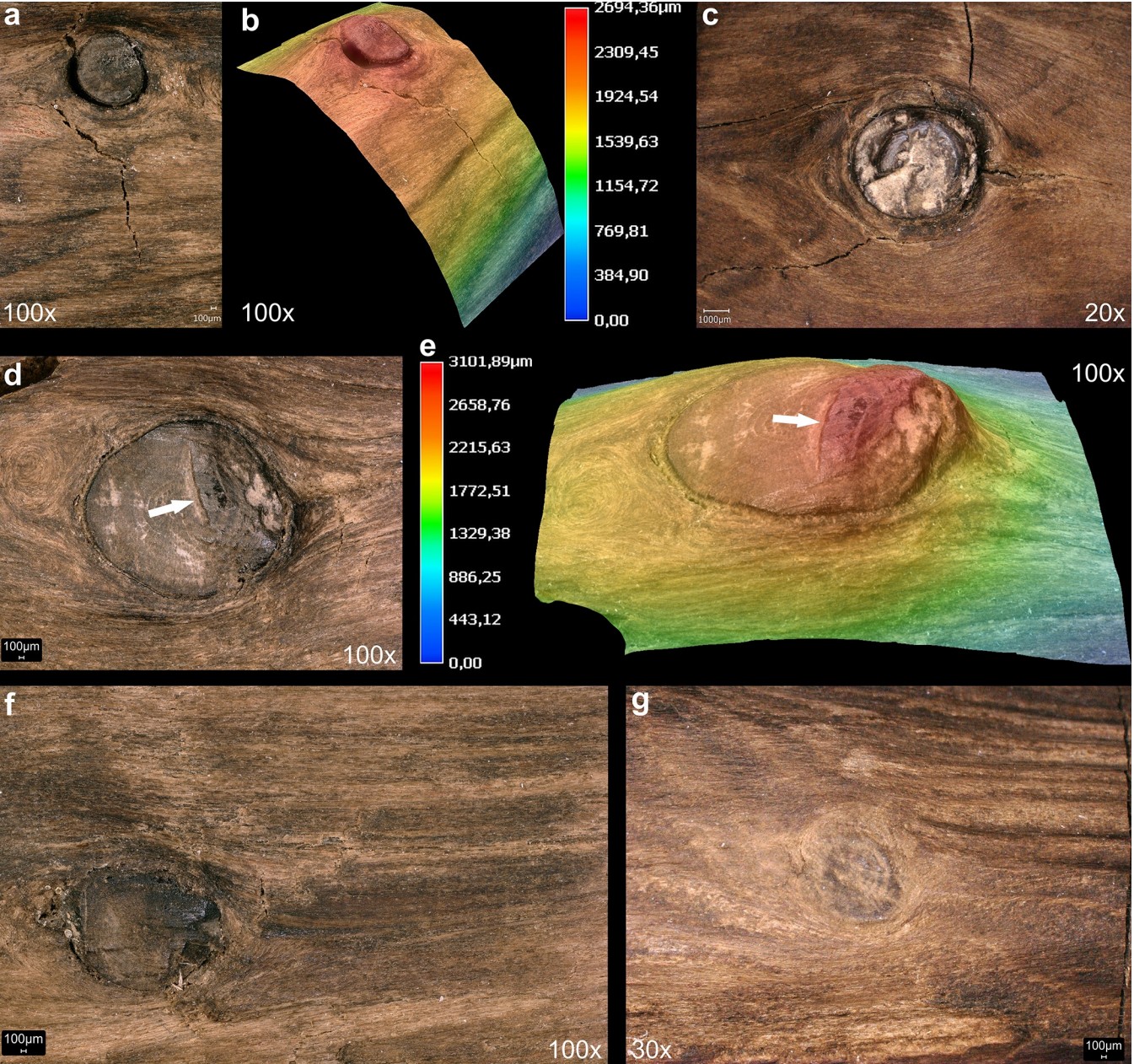

**Fig 15. Examples of worked knots.** Arrows on d and e point to a stop mark. Images T. Koddenberg.

Furthermore, the embedded resin varies in thickness which superimposed a background on all spectra of the wood. The unexposed wood component lignin is aromatic, and the embedded Kauramin resin adds aromaticity via its triazine ring [66]. Thus, the charring is not identifiable qualitatively by a new aromaticity peak from graphene (graphene aromaticity peak at for instance 1610 cm.1 [Factor 1990]) but instead only can be identified quantitatively. This also applies to oxidation-induced carbonyl functions, as both lignin [45] and Kauramin [66] introduce carbonyl functions. As the changes in IR spectra for the dark zones compared to light ones are quantitative rather than qualitative, the broad local variation of IR spectra on the

double-pointed stick did not allow us to differentiate chemically between visually light and dark regions.

### 3.3 Phase 2: Use, maintenance, and discard

The surface of the stick does not have evidence of impact damage or use fractures [cf. 8]. Surface damage on the second stick was interpreted as resulting from impact [8]. According to our analysis such damage is a result of taphonomic processes, and is discussed as such in the following section. An absence of irregular striations on the points makes it unlikely that the ends were used for activities such as digging, bark peeling or other extractive activities. There is a dark grey discoloration, with associated use polish on protruding knots (Fig 15C) in the midsection of the shaft of the tool (View A3–A4, ca 40–56 cm) where the stick was probably held. The association of polish and staining in this mid-section demonstrates that the tool had a long use life. In addition to the potential for use residues on the points (Section 3.2.2) this is the clearest evidence of use on the artefact. If the tool was multifunctionally used as a shorter stabbing weapon, the dark areas on the points could reflect residues such as blood or fat. Although some of the wooden tools from the site were likely repaired and maintained [7], there are no clear traces on this artefact that point to reworking after damage. The location of the double-pointed stick at some distance from other wooden tools, its completeness upon discard, and taphonomic evidence of rapid burial in mud (see 3.4) suggest that it was lost along the lakeshore during use rather than discarded due to breakage or cached for future use [15].

### 3.4 Phase 3: Taphonomy

The artefact was found along the former lakeshore. An absence of insect feeding traces, and of longitudinal cracks caused by desiccation and weathering, indicates that the tool was fully immersed in wet conditions shortly after discard, remaining in a wet context thereafter [12]. Bacterial decay consistent with anaerobic conditions is present. With no indication of distribution of taphonomic smoothing consistent with high-energy water conditions, the find was likely embedded within low-energy lakeshore sediments. In striking contrast to most of the surface of the tool that is very well-preserved, there are localised areas of relatively large, sharply demarcated surface damage, particularly on the excavated surface (View A1) and on the sides (Views A2 and A4), but not on the underside (View A3). This type of damage is characterised by sharp edges and flat bases, and their base often follows annual ring surfaces (Fig 16; see also Table S9 in S1 File, and S3 Dataset). Due to these characteristics, the fractures occurred after a breakdown of the cellulose and lignin that support the cell walls after being underwater for some time, and not as a result of impact when fresh [cf. 8]. Such fractures along annual growth rings have also been observed on recent natural wet wood that was walked on by humans (Fig S12 in S1 File). The quantity of this type of damage along the tool also rules out impact damage. Rather, the focus of this surface spalling along the top of the artefact suggests that it can best be attributed to trampling, a well-known taphonomic effect on bones, including on faunal remains from Schöningen [27].

An additional taphonomic signature on the artefact is discolouration, with a brownish-beige patina over all surfaces and features. On View A3 there is also flecking in the form of bright spots (Fig 17A–17C). These form a straight line along the sides (Views A2 and A4) and are not present on the excavation surface (View A1). We attribute this to fungal attack, specifically white rot which causes discolouration in the form of bleaching [e.g. 67]. Due to the positioning and focus of this discolouration, it occurred after discard but prior to total submersion.

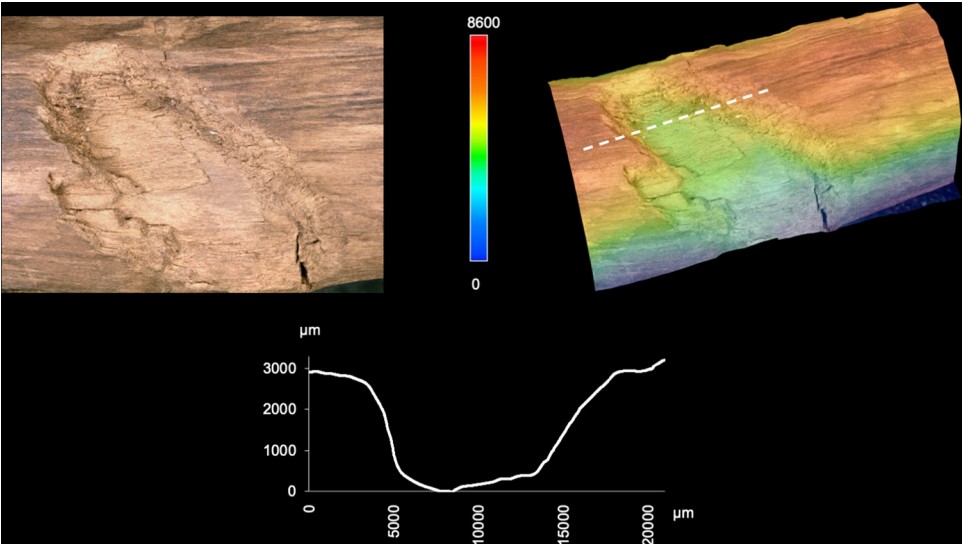

**Fig 16. 3D microscopy of an example of trampling damage.** View A1–A4 at 58.5–60 cm. Images T. Koddenberg.

The artefact bears some small (ca. 0.25 cm to 0.56 cm wide) deep, sharp-edged punctures (Fig S16 in S1 File; Table S7 in S1 File; S3 Dataset). 3D microscopy illustrates that these small punctures are often lenticular in plan and have straight edges (Fig S17 in S1 File). The damage is taphonomic as the wood would have to be very soft and degraded in order for such punctures to form. These punctures are typical for debarked surfaces of conifer wood from the Spear Horizon and can also be found on some fractured surfaces. In some instances they penetrate through the entirety of a fragment. These damages are attributed to root growth, probably from monocots, once the wood was very degraded.

Taphonomic compression had a small effect on the cross-section of the artefact, resulting in slight deformation in the direction from A1 to A4. This is evidenced by minor deformation of the rays, visible on prepared areas on the fracture surface. Some areas have irregular edges and wavy areas, also a result of minor taphonomic compression (Fig S11 in S1 File). Early documentation of the artefact after excavation shows that small transverse cracks were already present on the surface of View A1. These transverse cracks indicate stresses from sediment pressure, as well as some vertical deformation visible on Views A2 and A4. A small number of short longitudinal cracks are always associated with the transverse cracks, and are also related to taphonomic compression rather than weathering.

### 3.5 Phase 4: Excavation and post-excavation

There are transverse cracks along the piece (Fig S13 in S1 File), that occurred after conservation. There is also minimal damage to Point 1 including a small split that occurred after conservation (Fig S18 in S1 File). The artefact is currently in two fragments with a break located between 48.7 cm and 50.2 cm (Figs S13–S15 in S1 File). This can be best viewed on the 3D model, which can be accessed using the following link: Schöningen 13 II double pointed stick - 3D model by Denkmalatlas Niedersachsen (@denkmalatlas) [b3f2f12] (sketchfab.com). This post-depositional break is characterised by straight edges, with excavation images showing it was excavated and lifted in one piece. Archival records confirm that the complete fracture must have occurred sometime after 2015. However, the initiation of this break may have been

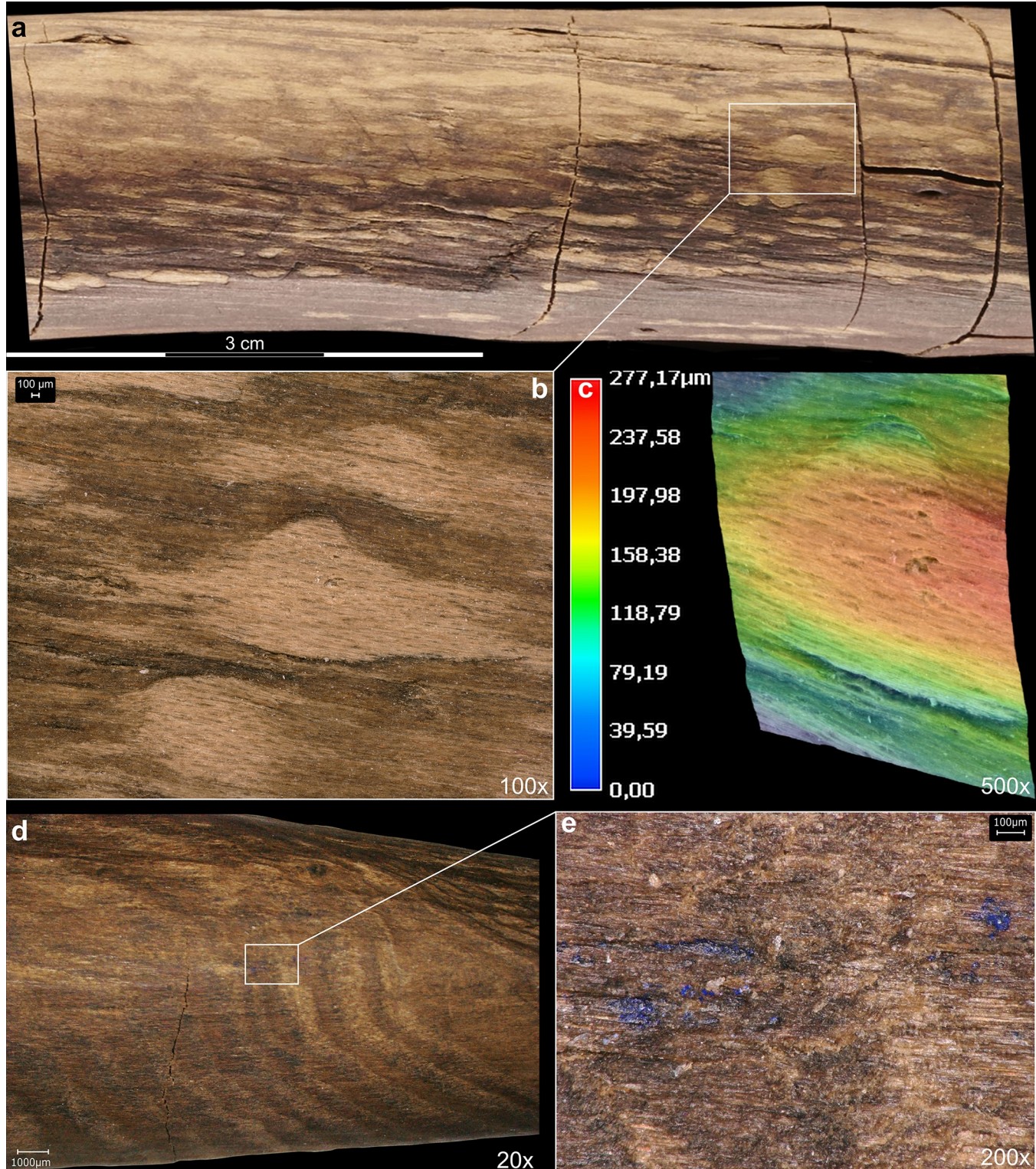

**Fig 17. Taphonomic and post-excavation features.** Flecking (a–b), and mineralisation (e) are visible. Images T. Koddenberg.

a crack from trampling damage, which is present at this location (Views A1–A4 and A1–A2; Fig 10). Blue mineralisation is present in the darker area on Point 1 (Fig 17D and 17E).

## 4. Discussion

### 4.1 Raw material selection and organisation

Human use of the lakeshore landscape of Schöningen 13 II-4 was during the very end of an interglacial, transitioning into a glacial period with cooling temperatures. In terms of raw material for the wood tools, the growth rings of the spruce used for spears and the throwing sticks indicate less favourable growing conditions, while in contrast the wider annual rings for the pine used for spear IV suggests a more favourable habitat [7, 19]. Like the spears, the tool was made using slow-grown spruce with extremely narrow tree rings and dense wood. Other Pleistocene wooden spears and wooden spear points were made of yew (*Taxus baccata*) [68–71], which was not available at Schöningen during that period [7, 19, 20]. The spruce wood used by the hominins for the production of weapons was considerably harder than spruce available in the region today. The selection for spruce may be because it was widely available and relatively easy to work in comparison with other available species, but also potentially reflects favourable properties for weapons [16, 72, 73]. Analyses of pollen and macro-remains suggests that the spruce wood did not grow at the lakeshore, and may have come from some distance away [16, 20]. For spruce, narrower annual rings can result from growth in poorer conditions, including high altitudes [e.g. 64]. One region where wood with such narrow rings could have been sourced was the forests of the neighbouring Elm hills, or the Harz Mountains approximately 40 km distance. In contrast to the spears, the blank selected was not a trunk, but a branch. This key difference means that it is unlikely to be a repurposed broken spear fragment. The hominins may have selected a branch for its natural bend which they then partially reduced through shaping. The use of a branch for this shorter tool could be an economic choice as it takes less time to remove a branch than to fell a tree. A smaller branch would also require less time to work down the material to arrive at the desired length and diameter.

### 4.2 Manufacture

After an initial rough stage of branch removal, debarking was an important next step in preparing the blank, even for the sections that were not further shaped. Debarking improves handling and flight performance, accelerates the drying process and thus hardening of the tool, and helps prevent insect attack during the use phase [74]. Like most of the Schöningen wooden tools, the double-pointed stick bears no traces of inner bark and/or cambium. The systematic and similar working strategy to shape both points resulted in the softer pith emerging along the sides. The shaping of the two points on the tool also resulted in retaining harder compression wood, something also remarked on for later archaeological wood weapons [75]. Working away the top of the branch, while leaving the compression wood along the underside could have various explanations. The most parsimonious one is that the underside of the branch with the compression wood is hardest to work, and therefore to ease the shaping process, more wood was removed from the softer top of the branch. However, whilst compression wood is harder, it is also more brittle which increases potential for breakage during use [76]. The retention of this harder but more brittle wood could yet have a functional advantage that is not yet understood, including improved penetration if used multifunctionally as a stabbing weapon in hunting or self-defence. The question of advantages and disadvantages of compression wood is one that future use experiments of wooden tools should address.

Regardless of the material properties of the compression wood, the shaping of the points resulted in a tool that retains a slight curvature. According to Bordes' [43] classification system,

completely straight throwing sticks require a circular cross-section for stability in flight. In contrast, those with a 'forme d'arc' or arc shape, consist of a gradual curvature with a constant diameter and a circular or slightly elliptical profile, and sometimes make use of a naturally curving branch [see also 77]. The Schöningen throwing stick appears to be somewhere between these straight and arced forms, with a moderate curvature, a natural elliptical cross-section of the shaft, and deliberate shaping of the ends into elliptical points (Figs 7 and 9; Fig S5 in S1 File). Deliberate removal of material in order to maximise or minimise curvature is also evidenced on ethnographic throwing sticks using X-Ray Tomography [77]. Pointed ends in association with the arced form create drag and slow rotation in flight, somewhat reducing the maximum effective distance [43].

Tool marks provide clear evidence that this tool was made using stone tools. In particular, tool facets with signature features leave a recognisable trace that capture the tool edge and working direction. The double-pointed stick also bears evidence of abrasion to round the knots and finish the points. Use-wear analysis on lithics from the same find horizon as the double-pointed stick indicates woodworking residues on both unretouched flint flakes and scraper tools [28]. With raw materials probably coming from some distance from the lake-shore, woodworking residues on flint tools at Schöningen may reflect curation including resharpening of damaged wooden weapons. A recent wider review found that in the Middle Palaeolithic bifaces and *pièces esquillées* were used for chopping wood, flakes and blades were utilised for shaping, carving, and working knots, while finishing work made use of scrapers and denticulates [78]. Later archaeological sites and ethnographic sources show that other materials such as shells and animal teeth can also be used as woodworking tools [79–82]. For the purpose of abrading wood, humans have made use of a wide variety of materials including tree bark, limestone, sandstone, pumice, handfuls of stone chips, river weed, and sea sponges [79, 83–85].

## 4.3 Use and function

Both points are intact, the shaft break is post-excavation, and there is an absence of diagnostic impact damage. There is a possibility that hominins cached weapons along the lakeshore, proposed previously for Schöningen [15]. Such a strategy was also suggested for archaeological remains of spears and boomerangs at the Holocene site of Wyrie Swamp (Australia) [86]. However, due to its location at some distance from other wooden tools, and the negative impact that moisture from wet storage would have on the integrity of the wood itself, we think this unlikely. Taking into account the tool's completeness, we conclude that the tool was also probably not discarded due to damage, but rather was lost during use.

Signs of discolouration and polish on rounded knots in the central section of the tool indicate where it was likely held, and suggest a long use life. There is an absence of traces on either point that would indicate use as a digging stick, characterised by use-wear focused at the ends including randomly oriented striations, blunt and split tips, pitting, splinter negatives, and use polish [87]. Use as a bark peeler is similarly unlikely not just because of an absence of wear at the points, but also because this artefact is beautifully worked and finely finished. As discussed above, the double-pointed stick could have had a secondary function as a short stabbing tool. In support of this are the finely worked points, shaped to retain harder compression wood.

The possibility of it functioning as a child's weapon is intriguing. Archaeological and ethnographic evidence from hunter-gatherer societies suggests children are provisioned with miniature versions of adult weapons that can be intended as toys for pretence play, learning tools, or as finely-made scaled-down weapons enabling them to participate in hunting [see 88 for a review]. However, while (like the spears) the maximum diameter is off-centre, unlike the

spears it is naturally curved. We have clearly described how the hominins worked the blank to partially straighten it, but the selection of a curved branch for a small child-sized spear seems unlikely. The longer spears, with the exception of Spear VI which has a kink, were generally straight prior to deposition [7], a key aerodynamic feature. While little is understood about spear hunting amongst Middle Pleistocene hominins, recent research shows that for contemporary foragers learning to hunt, the development of weapon skills begins in early childhood and intensifies in adolescence [89, 90]. However, the use of throwing sticks by children in recent societies supports the potential for such a tool to have been used at Schöningen by younger group members in games and/or communal hunting, whilst developing key hunting and throwing skills [43].

Several lines of evidence further support the original interpretation that this artefact, as well as the second shorter double-pointed stick, functioned as throwing sticks [2, 8]. Throwing sticks are non-returning projectile weapons thrown in a rotational motion around their centre of gravity [43, 91]. Our literature review (see S1 File, section 1.5 for methods, and S4 Dataset) provides morphometric data, with lengths ranging from 58.4 cm to 103 cm, diameters between 2.2 cm and 3.5 cm, and masses between 120 g and 265 g. Both double-pointed sticks from Schöningen fit within these ranges. Ethnographic throwing sticks are morphologically variable and include straight and curved shafts with circular, elliptical or more streamlined profiles (i.e., biconvex, plano-convex). Ends can be shaped to have elliptical or spherical profiles. One or both ends can be uniform to the shaft or be shaped into points. Some are expediently made with little or no modification.

Ethnographically, throwing sticks were used in various scenarios including in interpersonal violence, to kill pests, in self-defence against dangerous animals including snakes, and for hunting birds, small mammals, marsupials, and larger herbivores including duiker, reindeer and kangaroo (see S4 Dataset for sources; see also 91 for an additional review). The multifunctionality of throwing sticks as clubs and stabbing weapons is also discussed elsewhere [92–94]. Archaeological parallels include complete and fragmented wooden and mammoth ivory artefacts interpreted as throwing sticks from sites in North America [95, 96], Europe [97, 98], and Australia [99; see also 43, 91 for reviews]. Some archaeological throwing sticks are well-crafted tools while others were only minimally worked by debarking the ends [e.g. 96]. Both of the double-pointed sticks from Schöningen compare favourably with later archaeological examples and with ethnographic throwing sticks.

The two throwing sticks from Schöningen present a fascinating addition to the Middle Pleistocene toolkit, with new potentials in terms of hunting strategies, prey selection, and communal involvement in hunting including by children and/or adolescents. While it was previously thought that the capture of small, fast prey was a behaviour not seen until the Upper Palaeolithic [100] there is increasing faunal evidence of the exploitation of birds and lagomorphs at European sites from as early as MIS 11 [101, 102], albeit with an absence of evidence of the technologies with which they may have captured such animals. And because ethnographic evidence demonstrates that wooden spears are not just used to hunt large ungulates, but also for hunting small and fast terrestrial prey and for fishing [43] we must ask to what purpose would the Schöningen hominins so carefully and expertly craft these throwing sticks?

Middle Pleistocene and early Late Pleistocene hominins are often characterised as technologically limited by short-distance hunting technologies, with spears accurate only between 5–10 m away [103–105]. This model is challenged on the basis of ethnographic and experimental data of throwing spear use, including using replicas of Schöningen spears [36, 38, 41, 106, 107]. The throwing sticks at Schöningen point further to the use of medium distance hunting weapons. Determining accuracy distances of projectile tools presents challenges, as distances will vary depending on the skill of the thrower, throwing direction, size and

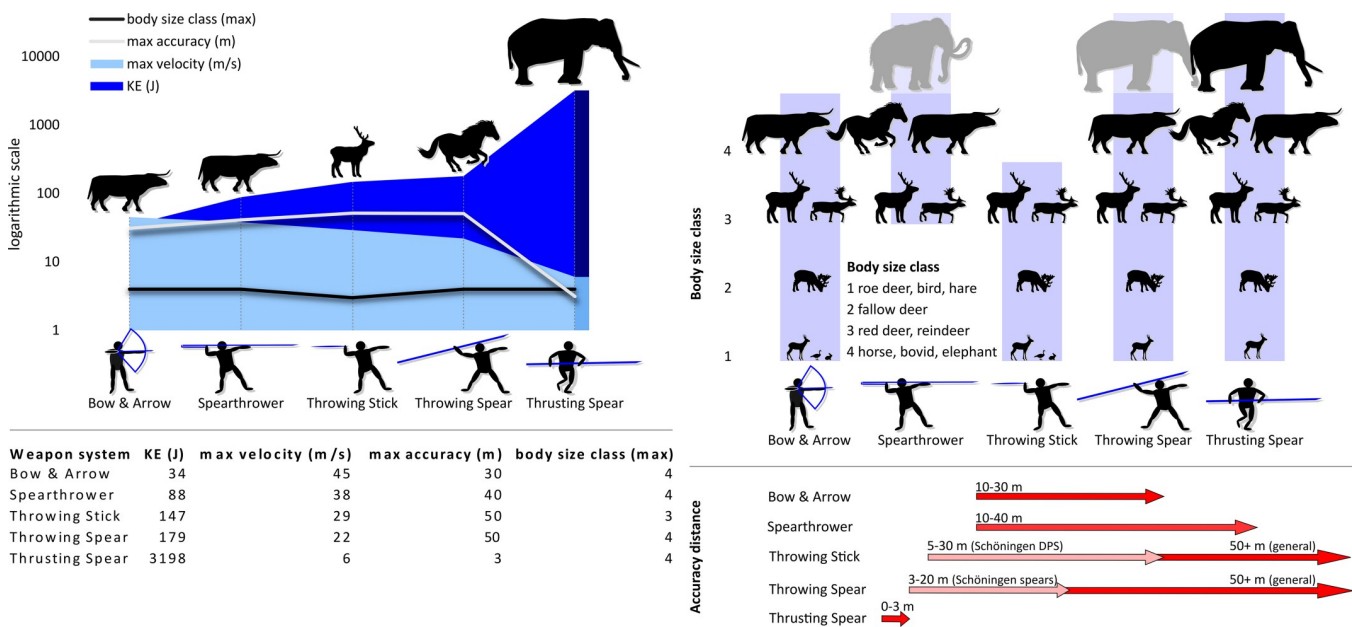

| Weapon system | KE (J) | max velocity (m/s) | max accuracy (m) | body size class (max) |
|---|---|---|---|---|
| Bow & Arrow | 34 | 45 | 30 | 4 |
| Spearthrower | 88 | 38 | 40 | 4 |
| Throwing Stick | 147 | 29 | 50 | 3 |
| Throwing Spear | 179 | 22 | 50 | 4 |
| Thrusting Spear | 3198 | 6 | 3 | 4 |

**Fig 18. Estimated performance parameters of five key Pleistocene weapons, and prey for which they are suitable.** Sources of estimated performance data: [36–38, 41, 43, 94, 103, 108, 110–123]. See S1 File, section 1.9 for further elaboration of sources for estimates. Diagram by D. Leder and A. Milks.

behaviour of prey, landscape, and weapon design. Furthermore, there are few experimental and ethnographic data on these tools. An experiment working with Daasanach pastoralists demonstrated throwing distances of up to 110 m using bent throwing sticks [108]. Noetling [94] noted that Aboriginal Tasmanian throwing sticks of similar morphometrics to the Schöningen sticks (but lacking a similar aerodynamic curve) were thrown at distances of 37 m. Bordes [43] also notes that throwing sticks with an elliptical cross-section, such as that seen on this double-pointed stick, can exceed 50 m, although accuracy distances can vary depending on multiple morphometric factors. Shorter distances of ca. 6–10 m for these tools to hunt small prey such as rabbits are also recorded [109]. Based on these ethnographic data, and prey behaviours of larger herbivores such as those known to have been hunted at Schöningen, we propose an estimated maximum accuracy distance for the Schöningen throwing sticks to be ca. 30 m [43, 94] (Fig 18; See S1 File, section 1.9 for further discussion and sources).

Experiments with throwing sticks recorded relatively high velocities and impact energies [108] (Fig 18; See also S1 File). Such tools could be used at both short and medium distances for stunning terrestrial and aerial prey, and/or breaking the legs of larger ungulates. In sum, the Schöningen throwing sticks, found in the same horizon as thrusting and throwing spears, demonstrate that the hominins were crafting a variety of differently designed high-energy weapons (Fig 18). In comparison with the Schöningen spears, the throwing sticks are lighter projectiles that are suitable for hunting a variety of prey types, and are potentially easier to use. Such features may therefore point to wider community involvement in hunting. Once again, the extraordinary preservation conditions at Schöningen enable us to better understand the range of sophisticated, variable and effective technologies and behaviour of Middle Pleistocene hominins.

## 4.4 Taphonomic effects on preservation of traces

Taphonomic factors affected the recognition of some manufacturing traces on the tool. However, significant surface damage is constrained to specific areas. The presence of similar

damage on other Schöningen wood artefacts suggests that it was likely trampled on by hoofed animals and/or humans in a post-depositional phase, when the wood was very soft. Further small areas of damage support our argument that the discard of the tool took place within a living wet lakeshore environment with the presence of growing plants and fungi. The effects of sediment pressure are on this piece relatively slight. Work to better understand the effects of taphonomy on the preservation of manufacturing and use traces on wood is ongoing.

## 5. Conclusion

Our detailed analysis of the double-pointed stick leaves no doubt that this was a well-planned, expertly manufactured, and finely-finished tool. The sequence of manufacturing stages appears to have been the removal of the branch from the tree, followed by rough removal of the branches. After oblique cuts were introduced to facilitate bark stripping, subsequent scraping was employed to fully debark the surface. Working down the knots and branch attachments likely took place after debarking, followed by shaping of regular points, partially straightening the natural bend. The final step was to abrade the surface to improve handling and/or performance, possibly alongside controlled seasoning to avoid drying cracks and warping, and to harden the wood. The fine surface, carefully shaped points and use polish suggests this was a piece of personal kit with repeated use, rather than an expediently made and discarded tool. The Schöningen hominins thus had the capacity for remarkable planning depth, knowledge of raw materials, and considerable woodworking skill, resulting in an expertly designed tool [124].

At Schöningen 13 II-4 at least 12 wooden hunting weapons were found on a ca. 120 m long preserved segment of a lake shore. In contrast to earlier ideas, the find situation can best be explained by repeated hunts rather than a single episode [125]. The double-pointed sticks were potentially used to assist the hunting of larger prey, but may have also been used for hunting birds and/or small mammals. Although there is to date no definitive evidence for the exploitation of small prey at Schöningen, both throwing sticks were found far removed from the spears and each in relatively isolated locations, hinting at the potential that they may have hunted small and fast prey. The fact that the throwing stick analysed here was in one piece on discard suggests they may have been lost amongst the lakeshore reeds.

Our results show the benefits of a systematic and methodologically sound approach to the study of these important early objects. New protocols and methods to recognise micro-traces decreases observational bias, in turn enabling recognition of anthropologically modified material in absence of tools or wood fragments with a clear morphology. These new techniques, concurrent with the development of expertise in the analysis of Pleistocene wooden finds, move forward our ability to understand the cultural biographies of Pleistocene technologies and behaviours. Specifically, the application of 3D microscopy to surface features and the use of micro-CT scanning enables a detailed analysis of raw material features and manufacturing techniques employed by the humans crafting these early, supposedly simple tools. The combination of techniques applied to this single tool demonstrates the depth of understanding that can be gained from 'zooming in' on single objects from archaeological sites with exceptional preservation.

## Supporting information

**S1 File. Further details on methods and results on the double-pointed stick (ID 1779).** (DOCX)

**S1 Fig. Perspective photograph of the double-pointed stick (ID 1779).** Photo: Volker Minkus.
(TIF)

**S1 Dataset. Annual ring analysis of the double-pointed stick (ID 1779).** Data collected by M.S.
(XLSX)

**S2 Dataset. Measurement data of the double-pointed stick (ID 1779).** Data collected by A. M. and J.L.
(CSV)

**S3 Dataset. 3D microscopy morphometric data of the double-pointed stick (ID 1778).** Data collected by T.K.
(XLSX)

**S4 Dataset. Ethnographic review.** Data collected by A.M.
(CSV)

## Acknowledgments

We would like to thank Luc Bordes and an anonymous reviewer for the constructive suggestions to improve the paper. We thank Waygate Technologies for the micro-CT scanning and Bernhard Schartel and Yannik Wägner at the Bundesanstalt für Materialforschung und -prüfung for their contributions towards the FTIR analysis. Many thanks to GOM Metrology for the 3D scanning of the object and to Hartmut Thieme and Werner Schoch for previous work on the Schöningen wood artefacts. We are grateful to Anna-Laura Krogmeier and all the staff at the Forschungsmuseum Schöningen for facilitating access to the artefact. Finally, we are grateful to Paula Garcia Medrano for her assistance with the 3D model.

## Author Contributions

**Conceptualization:** Dirk Leder, Thomas Terberger.

**Data curation:** Annemieke Milks, Jens Lehmann, Dirk Leder, Tim Koddenberg, Utz Böhner.

**Formal analysis:** Annemieke Milks, Jens Lehmann, Dirk Leder, Michael Sietz, Tim Koddenberg, Volker Wachtendorf.

**Investigation:** Annemieke Milks, Jens Lehmann, Dirk Leder, Volker Wachtendorf, Thomas Terberger.

**Methodology:** Annemieke Milks, Jens Lehmann, Dirk Leder, Michael Sietz, Tim Koddenberg, Utz Böhner, Volker Wachtendorf, Thomas Terberger.

**Project administration:** Annemieke Milks, Thomas Terberger.

**Resources:** Tim Koddenberg.

**Software:** Tim Koddenberg.

**Supervision:** Thomas Terberger.

**Visualization:** Jens Lehmann, Dirk Leder, Michael Sietz, Tim Koddenberg, Utz Böhner, Volker Wachtendorf.

**Writing – original draft:** Annemieke Milks, Jens Lehmann, Michael Sietz, Tim Koddenberg, Volker Wachtendorf, Thomas Terberger.

**Writing – review & editing:** Annemieke Milks, Jens Lehmann, Dirk Leder, Michael Sietz, Tim Koddenberg, Utz Böhner, Thomas Terberger.

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
