## [Decision Letter · Decision Letter 0]

20 Mar 2023

PONE-D-22-35734Perfectly shaped — A multianalytical study of a double-pointed wooden stick from Schöningen, GermanyPLOS ONE

Dear Dr. Milks,

Thank you for submitting your manuscript to PLOS ONE. After careful consideration, we feel that it has merit but does not fully meet PLOS ONE’s publication criteria as it currently stands. Therefore, we invite you to submit a revised version of the manuscript that addresses the points raised during the review process.

We look forward to receiving your revised manuscript.

Kind regards,

Marco Peresani

Academic Editor

PLOS ONE

Journal Requirements:

2. In your manuscript, please provide additional information regarding the specimens used in your study. Ensure that you have reported specimen numbers and complete repository information, including museum name and geographic location.

For more information on PLOS ONE's requirements for paleontology and archaeology research, see https://journals.plos.org/plosone/s/submission-guidelines#loc-paleontology-and-archaeology-research.

“The study was supported by the Deutsche Forschungsgemeinschaft (DFG, German Research Foundation) – project number 447423357. AM is funded by a British Academy Postdoctoral Fellowship (PF21/210027).”

“T.T. and this project is funded by the Deutsche Forschungsgemeinschaft (DFG, German Research Foundation) – project number 447423357.

https://www.dfg.de/

A.M. Is funded by the British Academy Postdoctoral Fellowship PF21/210027.

https://www.thebritishacademy.ac.uk/

The funders had no role in the study design, data collection and analysis, decision to publish, or preparation of the manuscript.”

Reviewers' comments:

Reviewer's Responses to Questions

**Comments to the Author**

1. Is the manuscript technically sound, and do the data support the conclusions?

Reviewer #1: Yes

Reviewer #2: Yes

2. Has the statistical analysis been performed appropriately and rigorously? 

Reviewer #1: Yes

Reviewer #2: Yes

3. Have the authors made all data underlying the findings in their manuscript fully available?

Reviewer #1: Yes

Reviewer #2: Yes

4. Is the manuscript presented in an intelligible fashion and written in standard English?

Reviewer #1: Yes

Reviewer #2: Yes

5. Review Comments to the Author

Reviewer #1: This paper is of great interest for the study of ancient wooden artefacts. It is well-written and well-organized. The recent publication of a second short double pointed stick on the site of Schoningen (Conard et al., 2020), including a full analysis, is pointing out the lack of a complete study of the first artefact of this type found in 1994. The present work is filling this gap, using microscopy CT to show in detail how the discovered stick is a result of a careful human wood shaping, how its curve had been slightly accentuated by working outside extremities, and how its has more probability to have been used as a throwing stick. This last point being quite important in regards of the functionality of the second shorter artefact found in 2011. Indeed, the present studied wooden artefact, being longer, one could think that it could be more adapted to be used for digging. ATR Infrared analysis, even if proved not successful in this case, is also complementing nicely the morphological and usewear study and worth to be reported.

Consequently, this article is worth to be published, but I think that two minor revisions are needed for its publication:

1 Archaic throwing stick having weak streamlined section (circular or elliptic) are mainly driven in their flight by initial throwing energy and by their mass, so this last key parameter should be estimated in the analysis. Authors have well discussed of the superior density of spruce wood at the period of manufacture compared to nowadays and be able to propose a range of initial weight value with estimation of the artefact volume by micro-CT, in relation with the actual weight of the preserved archaeological artefact.

2 I thinks authors should lower the estimated maximum range (50m) of the 1994 double pointed stick used as a throwing stick (both in text and figure 18). Indeed, if the reference (35) is reporting that throwing stick having circular and elliptic section can potentially reach this distance, others important parameters as weight (see revision 1) and stabilisation from curvature, can drastically affect their maximum range. If It’s true that it have been observed that Daassnatch throwing sticks can reach distance over 100 m (Roach and Richmond, 2015), in that case the type of throwing sticks used are more streamlined (elliptic to biconvex), highly stabilized by curvature (L shaped), more asymmetric (conservation of inertia momentum by additional weight on proximal blade), and probably made of a denser African wood than spruce tree. Additionally, as an experienced thrower, I have already tested two close replica of Schoningen 1994 double pointed stick in spruce as throwing stick and reach distances between 27-30 m, depending of the throwing technique (average of multiple throws in different directions relative to the wind). Consequently, in absence of throwing experiment included this study, I strongly suggest to lower this maximum range estimation. The maximum range need also to be distinguished from the useful hunting range which could be halved (15m). Yet, these projectiles are very useful on small animals and birds at very short range (i.e., Hopi people in Arizona were traditionally hunting rabbit between 6-10 m (Devereux, 1947). This can induce changes in the final discussion of the article too.

Additionally, I suggest others corrections in text and figures:

Text:

Abstract:

Last sentence:

Please specify which type of throwing sticks are long long-distance projectiles ? Artefact studied or other type ? Range overestimated. Please specify if you are speaking about maximum range or hunting useful range.

Page 4 line 22:

Reference bug “Error! Reference source not found”. Same bug can found in multiple occurrences further in the manuscript.

Page 17 discussion 4.1:

As wood density is one of the key factor for throwing stick performance, it would have been nice to indicate a range of value for spruce and yew density (for low growing tree in cold condition to fast growing tree in warmer condition)

4.3 use and function, page 19, line 34:

I suggest: circular, elliptical or more streamlined profiles (i.e., biconvex, plano-convex)

Page 20 line 26: I suggest replacing “morphology” by “characteristics” (morphology, section, length (or wingspan), curvature and weight) as these factors influence their performance. As already stressed in revision point 1, range of 50 m is overestimated, taking in account all characteristic or need to be fully justified according any experimental throwing data for the Schoningen double pointed stick studied here.

Figures:

Figures 5, 7, 9, 11: lacking scale, please add them.

Figure 8: Could have been interesting to combine this figure with the dotted limit of the tree branch (both in plan and section) given at the bottom of figure 4 to show clearly the removed wood part to accentuate curvature at the extremity. Removal of wood outside the curvature near extremity and inside the elbow is observed among manufacture of Australian throwing stick, see (Bordes, 2019). Additionally, that figure cloud may be be adjusted bigger to better visualization of CT sections.

Figure 18: I think that this type of diagram is may be too generalist, and don’t take in account the variety of weapon in a given class. So I’m not sure that it is useful to provide it in this article. The efficient hunting range for throwing stick will be highly dependent of the type considered (which have evolved during the whole Pleistocene). The same thing is true for the spear thrower dependent of the type of spear propelled as a light fast reed spear won’t go a the same distance and won’t have the same energy as a heavy long spear used to bring down a kangaroo in Australia. If you want to keep this figure anyway, please indicate in the right part if the weapon range indicated on this diagram are hunting efficient or maximum distance.

Suggested additional bibliography:

Bordes, L. (2019). X-Ray Tomography and Infrared Spectrometry for the Analysis of Throwing Sticks & Boomerangs. Journal EXARC - Experimental Archaeology, 3

Devereux, G. (1946). La chasse collective au lapin chez les Hopi, Oraibi, Arizona . Journal de la Société des Américanistes de Paris, 33, pp. 63-90.

Roach, N. T. & Richmond, B.G. (2015). Clavicle length, throwing performance and the reconstruction of the Homo erectus shoulder. Journal of Human Evolution, 80, pp. 107-113.

Reviewer #2: The paper presents the result of original research on a significant example of a wooden throwing stick from one of the most outstanding middle Pleistocene archaeological sites. The study uses cutting-edge methodology applied to unravel the biography of the object. Taking into account the underrepresentation of works dealing with prehistoric wood artefacts the methodology applied could be a referent for the study of similar artefacts. The work proposes a standardized protocol to carry out the study of wood artefacts very useful for undertaking future works.

The results reported here have not been published previously.

The paper is well structured and written, the methodology is clearly detailed in the supplementary materials and the results are well presented. Data availability is assured, it is provided as supplementary materials. Discussion of data is clearly performed and the data support the final conclusions. Conclusions are very relevant and provide new insight into hunting strategies and woodworking in the Middle Pleistocene.

See some minor comments in the text.

6. PLOS authors have the option to publish the peer review history of their article (what does this mean?). If published, this will include your full peer review and any attached files.

Reviewer #1: **Yes: **Luc Bordes

Reviewer #2: No

---

## [Author Response · Author response to Decision Letter 0]

2 Jun 2023

Response to reviewers for A double-pointed wooden throwing stick from Schöningen, Germany: results and new insights from a multianalytical study 

Many thanks for the useful comments from reviewers. We respond to each of these queries and suggestions in turn below. 

Editorial Comments

 and

2. In your manuscript, please provide additional information regarding the specimens used in your study. Ensure that you have reported specimen numbers and complete repository information, including museum name and geographic location.

The ID number was only in the figure, it has now been added to the text, the museum name was already in the text, but the geographic location has now been added as well. 

For more information on PLOS ONE's requirements for paleontology and archaeology research, see https://journals.plos.org/plosone/s/submission-guidelines#loc-paleontology-and-archaeology-research.

No permits were required, this sentence has been added to section 1.1.

“The study was supported by the Deutsche Forschungsgemeinschaft (DFG, German Research Foundation) – project number 447423357. AM is funded by a British Academy Postdoctoral Fellowship (PF21/210027).”

“T.T. and this project is funded by the Deutsche Forschungsgemeinschaft (DFG, German Research Foundation) – project number 447423357.

https://www.dfg.de/

A.M. Is funded by the British Academy Postdoctoral Fellowship PF21/210027.

https://www.thebritishacademy.ac.uk/

The funders had no role in the study design, data collection and analysis, decision to publish, or preparation of the manuscript.”

We have removed the funding information in the manuscript. We have added the following funding information:

The project is further funded by the Lower Saxony Ministry for Science and culture with funds from the Future Lower Saxony programme of the Volkswagen Foundation.

We have added this information to the cover letter. 

We are now including all the datasets as supplementary files (S1 Dataset through S4 Dataset). The model is available on the NLD website, and the link is embedded in the manuscript and S1 Supplementary Information file. 

We have added suggested references, as well as new references that strengthen arguments throughout. We list these references in our responses, as well as at the end of this response to reviewers. To our knowledge there are no retracted articles cited. 

Reviewer 1:

This paper is of great interest for the study of ancient wooden artefacts. It is well-written and well-organized. The recent publication of a second short double pointed stick on the site of Schoningen (Conard et al., 2020), including a full analysis, is pointing out the lack of a complete study of the first artefact of this type found in 1994. The present work is filling this gap, using microscopy CT to show in detail how the discovered stick is a result of a careful human wood shaping, how its curve had been slightly accentuated by working outside extremities, and how its has more probability to have been used as a throwing stick. This last point being quite important in regards of the functionality of the second shorter artefact found in 2011. Indeed, the present studied wooden artefact, being longer, one could think that it could be more adapted to be used for digging. ATR Infrared analysis, even if proved not successful in this case, is also complementing nicely the morphological and usewear study and worth to be reported.

Consequently, this article is worth to be published, but I think that two minor revisions are needed for its publication:

1 Archaic throwing stick having weak streamlined section (circular or elliptic) are mainly driven in their flight by initial throwing energy and by their mass, so this last key parameter should be estimated in the analysis. Authors have well discussed of the superior density of spruce wood at the period of manufacture compared to nowadays and be able to propose a range of initial weight value with estimation of the artefact volume by micro-CT, in relation with the actual weight of the preserved archaeological artefact.

Thank you, this was a really good suggestion that adds to the interpretation of the object. The actual mass of the object today, either before or after conservation is not relevant, because it was first waterlogged and then conserved with additional material. These processes affect the mass so much so that we do not provide these data as they are not relevant and may even be misleading. However, estimating the original mass makes a lot of sense. We have followed the reviewer’s suggestion and calculated the volume of the object using the scans, and estimated the mass on the basis of the upper density limits of modern day Norway spruce. This has been added to the results as follows:

The total volume of the two fragments together equals 239 cm3. Depending on the growing conditions such as climate, modern Norway spruce (Picea abies) can have air-dry density values ranging from 0.232 g/cm3 to 0.588 g/cm3 and a mean value of 0.344 g/cm3 (n=368) (63). Spruce growing in today’s modern temperate conditions tend to have wider annual rings and lower density than the double-pointed stick. For example, a sample with a mean annual ring width of 1.44 mm corresponded to an air-dry density of 0.460 g/cm3 (64). With annual rings considerably narrower on the double-pointed stick (mean = 0.2 mm), this would likely result in it having an higher density compared with modern spruce (65). On the basis of the upper limit of modern spruce density and considering the determined volume, we estimate the original mass of the tool to have been ca. 141 g. 

We also added the method for this calculation to section 2.1:

The volume data of the two fragments were calculated from the 3D model using Artec Studio 16 3D Software and Geomagic® Essentials.

2 I thinks authors should lower the estimated maximum range (50m) of the 1994 double pointed stick used as a throwing stick (both in text and figure 18). Indeed, if the reference (35) is reporting that throwing stick having circular and elliptic section can potentially reach this distance, others important parameters as weight (see revision 1) and stabilisation from curvature, can drastically affect their maximum range. If It’s true that it have been observed that Daassnatch throwing sticks can reach distance over 100 m (Roach and Richmond, 2015), in that case the type of throwing sticks used are more streamlined (elliptic to biconvex), highly stabilized by curvature (L shaped), more asymmetric (conservation of inertia momentum by additional weight on proximal blade), and probably made of a denser African wood than spruce tree. Additionally, as an experienced thrower, I have already tested two close replica of Schoningen 1994 double pointed stick in spruce as throwing stick and reach distances between 27-30 m, depending of the throwing technique (average of multiple throws in different directions relative to the wind). Consequently, in absence of throwing experiment included this study, I strongly suggest to lower this maximum range estimation. The maximum range need also to be distinguished from the useful hunting range which could be halved (15m). Yet, these projectiles are very useful on small animals and birds at very short range (i.e., Hopi people in Arizona were traditionally hunting rabbit between 6-10 m (Devereux, 1947). This can induce changes in the final discussion of the article too.

Thank you for these thoughts and suggestion to lower the accuracy range. We actually based this estimated range of 50 metres from the reviewer’s own work (Bordes 2014, p. 58), which states that for elliptical cross-sections (translated):

‘It is a circular section from which material has been removed on two opposite sides. It is therefore the most archaic profiling that can be found. The rotation speed is improved by a stronger penetration in the air with this profile compared to the previous one. Throwing sticks that are too narrow to develop a true biconvex profile may possess this profile. The elliptical section can make it possible to produce throwing sticks which reach a respectable distance which can exceed 50 meters, with a minimum removal of material..’

However, we appreciate that the potential throwing distances rely upon more than just the cross-section, also on length, mass etc. 

In our ethnographic literature search we found little in the way of estimated distances for throwing sticks of this size and morphology. There is also an absence of good quality published data of throwing sticks of this kind with highly skilled throwers. This has also been a major issue for throwing spears (javelins). Nevertheless, we found that Noetling 1911 gives descriptions of Aboriginal Tasmanian throwing sticks of similar morphometrics (but without the slight curve, which as we understand it can improve distance based on Bordes’ work) with eyewitness accounts of throwing these sticks described as follows: “It can be thrown with ease forty yards, and in its progress through the air goes horizontally, describing the same kind of circular motion that the boomerang does, with the like whirring noise.” 40 yards is equivalent to 37 metres. This makes sense in terms of accuracy distances, as many accounts of Aboriginal Tasmanian throwing spears record accuracy distances of >50 metres. 

It is also important to add that in our own, and also in the Hrncir review paper that has just been published, there are sources indicating the use of throwing sticks to hunt larger game too, including reindeer, kangaroo, duiker, etc. Hunting such prey, which are larger animals that are more difficult to approach, from 6-10 metres with a throwing stick is not likely to be successful, suggesting longer hunting distances. At least in the case of the kangaroo this would have been with sticks of a similar design to the those from our site (e.g. Noetling 1911). 

Given the reviewer’s account of experience throwing replicas (with the above caveats noted), and the ethnographic observations of similar sticks thrown by Aboriginal Tasmanians, we have reduced the distance estimates to 30 m in the text and Figure 18, and provided further sources for these, alongside a discussion of these sources in the SI. We hope this is an agreeable solution.

Manuscript:

Middle Pleistocene and early Late Pleistocene hominins are often characterised as technologically limited by short-distance hunting technologies, with spears accurate only between 5–10 m away (103–105). This model is challenged on the basis of ethnographic and experimental data of throwing spear use, including using replicas of Schöningen spears (36,38,41,106,107). The throwing sticks at Schöningen point further to the use of medium distance hunting weapons. Determining accuracy distances of projectile tools presents challenges, as distances will vary depending on the skill of the thrower, throwing direction, size and behaviour of prey, landscape, and weapon design. Furthermore, there are few experimental and ethnographic data on these tools. An experiment working with Daasanach pastoralists demonstrated throwing distances of up to 110 m using bent throwing sticks (108). Noetling (94) noted that Aboriginal Tasmanian throwing sticks of similar morphometrics to the Schöningen sticks (but lacking a similar aerodynamic curve) were thrown at distances of 37 m. Bordes (43) also notes that throwing sticks with an elliptical cross-section, such as that seen on this double-pointed stick, can exceed 50 m, although accuracy distances can vary depending on multiple morphometric factors. Shorter distances of ca. 6–10 m for these tools to hunt small prey such as rabbits are also recorded (109). Based on these ethnographic data, and prey behaviours of larger herbivores such as those known to have been hunted at Schöningen, we propose an estimated maximum accuracy distance for the Schöningen throwing sticks to be ca. 30 m (43,94) (Fig 18; See S1 Supporting Information, section 1.9 for further discussion and sources). 

And SI:

Estimates for the performance of prehistoric hunting weapons have been greatly bolstered in recent years through performance experiments involving experienced weapon users, reviews of ethnographic and ethnohistorical literature pertaining to similar weapon systems, and new ethnographic studies aimed at addressing such questions. We acknowledge that our data are limited, as contemporary experiences of Western hobbyists, athletes and archaeologists are unlikely to provide accurately replicate weapon use by people for whom subsistence technologies are socially embedded and learnt from early ages (21,22), and for which communities rely upon these technologies for survival (see also 23). Furthermore, ethnohistorical accounts were likely biased due to colonialist objectives and perspectives, and the extent to which ethnographic analogy is a useful tool for understanding the deep past is a matter of further concern (e.g. 24–26). 

With these caveats of the limitations of experimental and ethnographic data in mind, the following sources inform our diagram. For kinetic energy the following studies provided either direct calculations or paired velocity and mass data (13,14,27,28). The following studies provided experimental velocity data (13,14,27–29). Distance estimates are informed by experimental studies (13–15,30) alongside ethnohistorical and ethnographic accounts (31–41) and reviews (14 SI,19,42–44). Energies for wounding prey by body are after Tomka (44). 

Additionally, I suggest others corrections in text and figures:

Text:

Abstract:

Last sentence:

Please specify which type of throwing sticks are long long-distance projectiles ? Artefact studied or other type ? Range overestimated. Please specify if you are speaking about maximum range or hunting useful range.

In addition to reducing the maximum effective range throughout the paper, we have made this change to abstract, and rewritten much of the abstract as follows:

The site of Schöningen (Germany), dated to ca. 300,000 years ago, yielded the earliest large-scale record of humanly-made wooden tools. These include wooden spears and shorter double-pointed sticks, discovered in association with herbivores that were hunted and butchered along a lakeshore. Wooden tools have not been systematically analysed to the same standard as other Palaeolithic technologies, such as lithic or bone tools. Our multianalytical study includes micro-CT scanning, 3-dimensional microscopy, and Fourier transform infrared spectroscopy, supporting a systematic technological and taphonomic analysis, thus setting a new standard for wooden tool analysis. In illustrating the biography of one of Schöningen’s double-pointed sticks, we demonstrate new human behaviours for this time period, including sophisticated woodworking techniques. The hominins selected a spruce branch which they then debarked and shaped into an aerodynamic and ergonomic tool. They likely seasoned the wood to avoid cracking and warping. After a long period of use, it was probably lost while hunting, and was then rapidly buried in mud. Taphonomic alterations include damage from trampling, fungal attack, root damage and compression. Through our detailed analysis we show that Middle Pleistocene humans had a rich awareness of raw material properties, and possessed sophisticated woodworking skills. Alongside new detailed morphometrics of the object, an ethnographic review supports a primary function as a throwing stick for hunting, indicating potential hunting strategies and social contexts including for communal hunts involving children. The Schöningen throwing sticks may have been used to strategically disadvantage larger ungulates, potentially from distances of up to 30 metres. They also demonstrate that the hominins were technologically capable of capturing smaller fast prey and avian fauna, a behaviour evidenced at contemporaneous Middle Pleistocene archaeological sites. 

Page 4 line 22:

Reference bug “Error! Reference source not found”. Same bug can found in multiple occurrences further in the manuscript.

We have fixed this throughout (it was a linked reference to a Figure). 

Page 17 discussion 4.1:

As wood density is one of the key factor for throwing stick performance, it would have been nice to indicate a range of value for spruce and yew density (for low growing tree in cold condition to fast growing tree in warmer conditions.

We added the upper limits of wood density for contemporary spruce in the section where we estimate the mass. The nature of the growth rings are quite different however, and modern growing spruce may be a poor analogue. We are not sure why it makes sense to add density values for yew trees in this paper, as this wood is not present at 13 II-4. We have added a citation that does explore more the relationship between wood densities and weapon performance (Milks 2022), but there are many caveats including lack of clarity of densities of the wood used for the tools at Schöningen. For example in the Keunecke et al. 2009 paper cited the ‘narrow growth ring’ spruce analysed had a mean growth ring width of 1.4 mm, and a wood density of 0.463 g/cm3. The growth rings of the object analysed here are on average 0.2 mm. This is substantially different. It is a future aim to test possibilities of estimating the original wood density at the site. 

4.3 use and function, page 19, line 34:

I suggest: circular, elliptical or more streamlined profiles (i.e., biconvex, plano-convex)

We have made this change. 

Page 20 line 26: I suggest replacing “morphology” by “characteristics” (morphology, section, length (or wingspan), curvature and weight) as these factors influence their performance. 

We have changed ‘morphology’ to ‘characteristics’. 

As already stressed in revision point 1, range of 50 m is overestimated, taking in account all characteristic or need to be fully justified according any experimental throwing data for the Schoningen double pointed stick studied here.

As described above, we have explained where our original distance estimate originated from. We have provided further distance data for a similarly designed throwing stick (Aboriginal Tasmanian), and added in discussion of various caveats. In addition to the text as changed in the abstract and discussion, and estimates altered in Figure 18, clarifying what we mean for the site vs. for the weapon category as a whole, we have added further data for Figure 18 in the SI, in a new section. 

Figures:

Figures 5, 7, 9, 11: lacking scale, please add them

We have added scales to these figures. 

Figure 8: Could have been interesting to combine this figure with the dotted limit of the tree branch (both in plan and section) given at the bottom of figure 4 to show clearly the removed wood part to accentuate curvature at the extremity. Removal of wood outside the curvature near extremity and inside the elbow is observed among manufacture of Australian throwing stick, see (Bordes, 2019). Additionally, that figure cloud may be be adjusted bigger to better visualization of CT sections.

We have followed the latter suggestion and made the CT sections more visible at a higher resolution. The reader can compare these two figures (Figure 4 and Figure 8). 

Figure 18: I think that this type of diagram is may be too generalist, and don’t take in account the variety of weapon in a given class. So I’m not sure that it is useful to provide it in this article. The efficient hunting range for throwing stick will be highly dependent of the type considered (which have evolved during the whole Pleistocene). The same thing is true for the spear thrower dependent of the type of spear propelled as a light fast reed spear won’t go a the same distance and won’t have the same energy as a heavy long spear used to bring down a kangaroo in Australia. If you want to keep this figure anyway, please indicate in the right part if the weapon range indicated on this diagram are hunting efficient or maximum distance.

As far as we are aware, there are no good quality published experiments on accuracy rates for given distances of throwing sticks. As stated above, we have added further references, adjusted the maximum accuracy distance down to 30 m and made amendments in the text. We favour keeping the image for two reasons. First, it includes the throwing stick as an important Pleistocene technology, one that is absent in similar illustrations (e.g. Iovita et al. 2014; Stodiek 1991; Milks 2018). Second, it helps make another important point that the Schöningen hominins possessed projected tools, not just hand-held thrusting and stabbing tools. The functional purpose of such throwing sticks is important to illustrate. We state in the caption that these are estimates, and cite the sources for these estimates. It is also clear in the figure that these are considered as maximum, although that could change with further experimental data. 

The kinetic energy is calculated based on experimental data from Roach & Richmond (as cited). Although the Schöningen throwing sticks may or may not have been lighter (as we state in our mass estimate, it is full of caveats including the inability to accurately estimate the original wood density), lighter projectiles can be thrown at higher velocities, which has an exponential effect on kinetic energy. Until recent spear throwing experiments demonstrated otherwise (Milks et al. 2019; Coppe et al.) it was also thought that throwing spears, including the Schöningen spears were also ‘low velocity’ weapons. For now we will keep the maximum KE estimates as they are based upon experimental throwing, recording velocity with a radar gun. As such they are indeed lower than maximum KE for spears, but not be a whole lot, because of increased velocities. It should be clear in Fig 18 now that this for the category as a whole, not just at Schöningen. 

Suggested additional bibliography:

Bordes, L. (2019). X-Ray Tomography and Infrared Spectrometry for the Analysis of Throwing Sticks & Boomerangs. Journal EXARC - Experimental Archaeology, 3

Devereux, G. (1946). La chasse collective au lapin chez les Hopi, Oraibi, Arizona . Journal de la Société des Américanistes de Paris, 33, pp. 63-90.

Roach, N. T. & Richmond, B.G. (2015). Clavicle length, throwing performance and the reconstruction of the Homo erectus shoulder. Journal of Human Evolution, 80, pp. 107-113.

Thank you for these additional references. The Roach and Richmond paper was already included as a citation, we have further included it in the caption data for Figure 18 and explanation thereof in the SI. We have added the Devereux paper amidst the discussion of throwing distances (as discussed above). 

We have also added Bordes 2019 in relation to straightening the stick in the discussion (4.2). 

Deliberate removal of material in order to maximise or minimise curvature is also evidenced on ethnographic throwing sticks using X-Ray Tomography (77). 

Additional points from the pdf:

Reviewer: Luc Bordes

Provide more detail of morphological description (maybe in Supplementary material)

We have made this change:

Morphological descriptions, further details of which are found in S1 Supplementary Information, follow Bordes (43). 

And in the SI:

1.7 Morphological description

Morphological descriptions of the throwing stick follow methods and terms in Bordes (19). Specific to this object include morphological classification on the basis of profile (e.g. circular, oval, elliptical), type and symmetry of form (e.g. straight shape, curved shape with enlarged head, crescent), and end type (e.g. pointed, bevelled, rounded). 

Which are these phases?

Later you explain the phases are described in a table, include here this comment and the reference to the table

As we explain the phases later, we thought it clearer to just make the following alteration here:

All traces were mapped onto a digital drawing of the double-pointed stick.

And as a result of this reviewer’s comment we also realised that the phases should be briefly explained in the main text, not just in the Supplementary Information, so in the section 2.5 we made this alteration to clarify each phase:

The cultural biography of the artefact is explored through five phases from raw material sourcing (Phase 0), manufacture (Phase 1), use, maintenance and discard (Phase 2), taphonomy (Phase 3), and excavation and post-excavation (Phase 4) (see Table S3 in S1 Supporting Information for a more detailed description of each phase). 

Which points? The extremes?

Yes, we meant the two extremities, and have clarified as follows:

The two extremities are designated as Point 1 and Point 2 (Figure 1).

In the next paragraph there is a similar sentence

We have deleted the first sentence and kept the second. 

You suggested that bark was preserved for a period of time after removing the branch of the trunk, clarify how you recognize the bark was pulled fresh or after drying.

Thank you, we realise this was unclear. We have clarified that it is the presence of the oblique cut marks that likely reflects cutting into the bark to remove it in strips. 

The tool is fully debarked, with no outer bark, inner bark or cambium (Fig 10). A series of striations oriented obliquely to the shaft have profiles and lengths (ca. 5 mm to 15 mm) consistent with cut marks (Figs 11 and 12; Table S5 in S1 Supporting Information). These cut marks likely facilitated the debarking process, allowing the toolmaker to cut into the bark and pull it off in strips when relatively fresh. The longest cut mark is arc-shaped and is over 50 mm long (View A2–A3 from 14.5 cm to 19.5 cm; Fig S7 in S1 Supporting Information). Micro-analyses show this mark has fibre deformation and a profile consistent with an angled cut with a sharp tool edge (Fig 12). In general the 3D microscopy demonstrates morphometric variability (Table S9 in S1 Supporting Information; S3 Dataset). Organised groups of longitudinal parallel and sub-parallel striations are likely scraping marks (Fig 13) to remove any remaining bark tissue, and to regularise the surface of the tool.

We have also amended the text to reflect that the marks suggest that the piece was debarked and shaped first, and any seasoning occurred after this step. 

An absence of significant surface or internal drying cracks suggests the wood dried slowly and evenly. Cut wood loses its natural moisture until it is in equilibrium with the surrounding environment, and if freshly cut and debarked wood is allowed to dry too quickly it can develop significant cracks and can also warp (53). Seasoning to remove moisture at a uniform rate can be accomplished in different ways, including allowing a branch or trunk to dry slowly with the bark attached, storing the wood in such a way as to allow uniform air circulation, storing it with worked wood chips, or over a fire to dry the wood.

Which one? add some more information on tree species present in the surroundings

We have added the following information to the background section, including further references (also in response to reviewer 2): 

Tree species previously evidenced in the Spear Horizon include pine (Pinus), with dropping levels of birch (Betula), and very few alder (Alnus), willow (Salix), juniper (Juniperus) and spruce/larch (Picea / Larix) (16,19–21). Spruce pollen is sparse in the profile, and is thought to have originated from a significant distance to the lakeshore (19,20). 

And in Section 4.1

The double-pointed stick was manufactured using spruce (Picea sp.) (see also 2). The use of spruce is in keeping with the wider sample of worked wood from the find horizon (13-II 4b and 4c). Ongoing species idenfication the wood material by one of the authors (M.S.) as part of the current research project demonstrates that in this same find horizon hominins also exploited pine (Pinus sylvestris) and larch (Larix), alongside a background presence of willow (Salix) and/or poplar (Populus), species which do not show signs of human modification. According to palynological analyses, spruce did not belong to the natural background vegetation at the site and the raw material was introduced by hominins (16,19,20). 

Difficult to stablish the distance from the pollen data.

We agree, this is difficult to know for certain, we use the term ‘may have’ to show we are not definitely sure. The suggestion of the mountainous region is not about distance per se but rather that it is in these conditions where such slow growing wood could be found. We have amended the sentence with a relevant citation of such growing conditions to reflect this. We have further added references, including from the 2023 paper by Urban et al. in support of this argument that potentially some wood could have originated from the closer Elm hills, but likely farther distance (See also Urban et al. 2023), for example the Harz Mountains. 

For spruce, narrower annual rings can result from growth in poorer conditions, including high altitudes (e.g. 64). One region where wood with such narrow rings could have been sourced was the forests of the neighbouring Elm hills, or the Harz Mountains approximately 40 km distance.

Reviewer #2: 

The paper presents the result of original research on a significant example of a wooden throwing stick from one of the most outstanding middle Pleistocene archaeological sites. The study uses cutting-edge methodology applied to unravel the biography of the object. Taking into account the underrepresentation of works dealing with prehistoric wood artefacts the methodology applied could be a referent for the study of similar artefacts. The work proposes a standardized protocol to carry out the study of wood artefacts very useful for undertaking future works.

The results reported here have not been published previously.

The paper is well structured and written, the methodology is clearly detailed in the supplementary materials and the results are well presented. Data availability is assured, it is provided as supplementary materials. Discussion of data is clearly performed and the data support the final conclusions. Conclusions are very relevant and provide new insight into hunting strategies and woodworking in the Middle Pleistocene.

See some minor comments in the text.

6. PLOS authors have the option to publish the peer review history of their article (what does this mean?). If published, this will include your full peer review and any attached files.

Do you want your identity to be public for this peer review? For information about this choice, including consent withdrawal, please see our Privacy Policy.

Reviewer #1: Yes: Luc Bordes

Reviewer #2: No

Additional points from PDF Reviewer 2

The work of Bigga 2018 (Bigga, G., Die Pflanzen von Schöningen, Heidelberg: Propylaeum, 2020 (Forschungen zur Urgeschichte aus dem Tagebau Schöningen, Band 3) is not cited here.

These sentences right at the start reference only primary publications of discoveries of wooden artefacts (or in the case of the Shigir Idol, the redating and reanalysis which re-dated it to the Late Glacial). In that context Bigga’s PhD which looked more broadly at the plant remains at Schoningen, is not referenced here as it was Hartmut Thieme who discovered the wooden artefacts at Schöningen. However, we are pleased to add this citation, and Bigga’s JHE article in an extension of a sentence a few paragraphs later. Her work was also already represented, but has also been now added elsewhere in the paper. 

Subsequent excavations of sediment sequence 4 (Schöningen 13 II-4) yielded faunal remains including butchered animals and flint artefacts (2), while exceptional botanical preservation shows the extent of plant materials available to the Schöningen hominins for technological, subsistence and medicinal purposes (15,16).

In addition to Kolfschoten 2015, another detailed study is by Voormolen 2008 (Voormolen, B. Ancient hunters, Modern Butchers: Schöningen 13 II-4, a Killbutchery Site Dating from the Northwest European Lower Palaeolithic. University of Leiden; 2008).

Yes, good idea, we have now cited that, as well as his published version in Journal of Taphonomy. 

Correct: Homotherium (Look Lit- 21.) 

We corrected this. 

In addition or alternatively to 24 you can also cite: Serangeli J, Conard NJ. The behavioral and cultural stratigraphic contexts of the lithic assemblages from Schöningen. J Hum Evol. 2015, 89: 287-297.

We have added this citation. 

It should be “double-pointed“, as done in the rest of the manuscript.

We have fixed the instance where it was not hyphenated. 

Check size: Thieme 2008 reports 4.7 cm at Page 144, Tab. 4.

4.7 cm is what we already had in the text. Schoch et al. 2015 is the best up to date information on the spear measurements. The sentence reflects that these are previously published data. 

The complete spears are double-pointed, and published data with previous measurement techniques report them as being between ca. 184 cm and ca. 253 cm in length and between ca. 2.3 cm and 4.7 cm in maximum diameter (8).

Sector 2 is not described in the text. 

It is located ca. 16 metres away from the nearest spear, spear nr. I

In this sentence you can also cite 7 (Conard et al. 2020), at place or in addition to 16 (Böhner et al.2015) because this precedes the discovery of the second throwing stick.

We have made all three of these changes, the sentence now reads: 

It was the only wooden artefact in square meter x 684/ y 31, is located 16 metres away from the nearest spears (Fig 3) and ca. 120 m from the second double-pointed stick (square meter x 772/ y -49) (8,18). 

You should revise this sentence because Conard et al 2020 did not use terms like shorter thrusting or stabbing weapon, digging stick, bark peeler, or children’s spear. Conard et al 2020, supported the artefact was a throwing stick.

Some of these functions are discussed in Conard et al. (some are dismissed, others are acknowledged to be difficult to rule out). We have amended these sentences as follows to reflect clearly which additional/alternative functions have been suggested in which papers, rather than putting references all together at the end of the sentence. We added a few references of examples that support an interpretation of a primary function as a throwing stick. 

Both of the short double-pointed tools, measuring under 1 metre, have various functional interpretations, but are most often viewed as ‘throwing sticks’ (2,8,41–43). The artefact analysed here is hypothesised to have potential additional or alternative functions including as a short thrusting or stabbing weapon (2), a digging stick (7), a bark peeler (44), or a child’s spear (7). 

Figure 3: The North arrow must be adjusted. Compare with your Fig. 2 of the paper.

Look also Bigga 2018, Page 170, Abb. 54 c.

Thank you for pointing out this discrepancy, which was due to excavation grid orientation vs magnetic north. This has been fixed. 

Avoid pre-interpretation: eliminate manufacturing from this sentence.

We have removed this word.

Werner Schoch (in Thieme, 1997) first determined this wood as picea sp. It should be nice to mention that your determination confirmed the previous one.

We have not made a change here, as this is methods and it would not be the right place for this citation. The appropriate acknowledgement was already in the results section, 3.1:

The double-pointed stick was manufactured using spruce (Picea sp.) (see also 2). 

(We have now added ‘see also’ just before the reference number to make this point even clearer.)

We also added the citation of both Thieme 1997 and Schoch et al. 2015 in the introduction mentioning that most of the tools were made from spruce. 

This statement should be also referred to literature. By Urban et al. 2015 in 4 b (c), page 67 “Pinus increases, Betula drops, very few Alnus, Salix, Juniperus, Picea (Larix). 

Populus is not in 4b or 4c but in 4f. 

Bigga et al 2015 as well Schoch et al 2015 did not mention Populous.

This is a misunderstanding of data from previous research vs. the current project. We recognise that this was unclear, and have amended the sentences to reflect this. In the current project, one of the authors (M.S.) has been conducting microscopic analyses of all of the wood from the find horizon, including the background wood. These findings will be published in greater detail in subsequent publications, but for now we can clarify that in 13-II 4, including for 4b and 4c, the findings are that there is a presence of Salix and/or Populus. A microscopic distinction of willow and poplar is based only on the shape of the edge cells of the rays. When dealing with young branch wood, no distinction can be made with heterogeneous edge cells which are normally typical of willow. Because the sample consists mostly of branch wood, and/or the condition of the samples does not allow any further differentiation, most of the natural 13 II-4 woods examined so far has been classified as Salix / Populus. To be cautious, we have clarified that this indicates willow and/or poplar. The presence of willow/poplar has been confirmed in this ongoing analysis in layers 4b and 4c, even though it was not mentioned in Bigga et al. 2015 or Schoch et al. 2015. This new information will be published in more detail in forthcoming publications. We have now added a few sentences about the previous work regarding the tree species in the background to the site section 1.1:

Tree species previously evidenced in the Spear Horizon include pine (Pinus), with dropping levels of birch (Betula), and very few alder (Alnus), willow (Salix), juniper (Juniperus) and spruce/larch (Picea / Larix) (16,19–21). Spruce pollen is sparse in the profile, and is thought to have originated from a significant distance to the lakeshore (19,20). 

We have also amended the relevant section in 4.1 accordingly:

The double-pointed stick was manufactured using spruce (Picea sp.) (see also 2). The use of spruce is in keeping with the wider sample of worked wood from the find horizon (13-II 4b and 4c). Ongoing analysis of the wood material by one of the authors (M.S.) as part of the current research project demonstrates that in this same find horizon hominins also exploited pine (Pinus sylvestris) and larch (Larix), alongside a background presence of willow (Salix) and/or poplar (Populus), species which do not show signs of human modification. 

Insert closing bracket “ ) “

We have made this change

The scale below the throwing stick and that of the sections are markedly different. Insert scale of the sections.

Thank you, we have made this change, the CT sections now have their own appropriate scale. Furthermore, the CT slices are now of a better resolution. 

Figure 11: Misse scale?

Many thanks for spotting this. We have added a scale to Figure 11. 

Both throwing sticks, as well the burnt wooden artifact are located in similar context as the spears. Do you conclude therefore that all the spears were probably not discarded due to damage but lost during use? Bigga 2018, 165-166 proposes hypotheses about the storage of the wood artefacts.

Good point, we should discuss the discard here. The research on the wider collection is ongoing, and the results of this wider context in terms of discard etc the spears and other wooden artefacts will follow in subsequent publications. This paper is only addressing the double-pointed stick. They are located in a similar context to spears, but at some distance away. We are certainly considering all the scenarios, including storage of wood and tools at the lakeshore for the wider collection, and will include that also in future publications. We added this sentence to 3.3, citing Bigga’s idea of storage at the end. 

The location of the double-pointed stick at some distance from other wooden tools, as well as its completeness upon discard suggests that it was lost along the lakeshore during use rather than discarded due to breakage or cached for future use (15). 

In Section 4.3 we write:

There is a possibility that hominins cached weapons along the lakeshore, proposed previously for Schöningen (15). Such a strategy was also suggested for archaeological remains of spears and boomerangs at the Holocene site of Wyrie Swamp (Australia) (86). However, due to its location at some distance from other wooden tools, and the negative impact that moisture from wet storage would have on the integrity of the wood itself, we think this unlikely. Taking into account the tool’s completeness, we conclude that the tool was also probably not discarded due to damage, but rather was lost during use. 

It’s not necessary to put “H.” for Homo, if you use Homo on side 20 and 21

We changed H. to Homo

Until now, the discussion was in “4.3 Use and function” “short stabbing tool“, or may be “child’s weapon”. There is a change to the evidences for “the original interpretation that this artefact functioned as a throwing stick“ … 

An additional and important argument is the comparison with the second throwing stick from the ‘Spear Horizon South’, as written at the beginning of the paper: “Although it was the first significant find, it remained an anomaly until a second stick of similar size and shape was recently discovered in new excavations of the ‘Spear Horizon South’ (7).”

Such a reference here would help to understand, why you speak of 2 throwing sticks from now on. 

“Both double-pointed sticks from Schöningen fit within these ranges.” And later “The two throwing sticks from Schöningen present …“

We are not certain we understand the reviewer’s points here, but we think the request is to reference both sticks at the start of the paragraph. We have made the following change to that sentence, including adding the reference to Conard’s paper. 

Several lines of evidence further support the original interpretation that this artefact, as well as the second shorter double-pointed stick, functioned as throwing sticks (2,8).

A function as a throwing stick does not rule out a function as a child’s weapon, as is already in the text on the previous page. Ethnographically, children use throwing sticks, these can be tools that illustrate wider community involvement in hunting. We added this further in a subsequent paragraph:

The two throwing sticks from Schöningen present a fascinating addition to the Middle Pleistocene toolkit, with new potentials in terms of hunting strategies, prey selection, and communal involvement in hunting including by children.

Page 5 “at least ten spears“ + 2 throwing sticks = 12

We have made this change. 

In addition to those responses to reviewer comments above, we made some further changes to improve the manuscript and supplementary information. We changed the title, as ‘Perfectly shaped’ suggested the tool reflected an evolutionary peak in design, which it does not. In addition: 

• We merged the two fragments to make a single 3D model, and have uploaded it to the NLD’s Sketchfab account. We have added the link in the manuscript and the SI. 

Manuscript:

This can be best viewed on the 3D model, which can be accessed using the following link: Schöningen 13 II double pointed stick - 3D model by Denkmalatlas Niedersachsen (@denkmalatlas) [b3f2f12] (sketchfab.com)

SI:

The two fragments were merged using Blender. The model can be viewed and accessed using the following link:

Schöningen 13 II double pointed stick - 3D model by Denkmalatlas Niedersachsen (@denkmalatlas) [b3f2f12] (sketchfab.com)

• We added some further ethnographic information from both our own review and that of a recently published review of clubs and throwing sticks. 

Ethnographically, throwing sticks were used in various scenarios including in interpersonal violence, to kill pests, in self-defence against dangerous animals including snakes, and for hunting birds, small mammals, marsupials, and larger herbivores including duiker, reindeer and kangaroo (see S4 Dataset for sources; see also 91 for an additional review). The multifunctionality of throwing sticks as clubs and stabbing weapons is also discussed elsewhere (92–94).

• Figure 14 has been replaced with clearer example of a stop mark. 

• We expanded section 3.3 to reflect observations mentioned in the abstract and discussion, but elaborated upon in this section. 

• A table for Phase 2 was missing in the SI, and we have added this. 

• We had a further discussion about excavation damage to Point 2, and after consultation with pre-conservation images, we have decided that we are not convinced this is excavation damage, as the colour pre-conservation of this point and the surrounding wood is similar. Therefore we removed the first sentence of the results of excavation/post-excavation damage:

The find was discovered during a rescue excavation, and minor damage may have removed the tip of Point 2 (Fig 9).

• We added all of the data files as supplementary files to make them directly available with the article

• We added a perspective image as a supplementary file (S1 Fig.) which shows a different view of the artefact. 

• We added the links for the 3D model

• We added a new section to the SI detailing the sources for estimates, and going a bit more in depth into explanations for these estimates. 

• We added following references to the main manuscript to improve the paper, as a result of suggestions of the reviewers, and where further and/or new publications supported our article. These additions include:

16. Bigga G, Schoch WH, Urban B. Paleoenvironment and possibilities of plant exploitation in the Middle Pleistocene of Schöningen (Germany). Insights from botanical macro-remains and pollen. Journal of Human Evolution. 2015 Dec;89:92–104. Available from: http://linkinghub.elsevier.com/retrieve/pii/S0047248415002456

20. Urban B, Kasper T, Krahn KJ, Kolfschoten T van, Rech B, Holzheu M, et al. Landscape dynamics and chronological refinement of the Middle Pleistocene Reinsdorf Sequence of Schöningen, NW Germany. Quaternary Research [Internet]. 2023 Apr 18 [cited 2023 May 26];1–30.

24. Voormolen B. Ancient hunters, modern butchers: Schöningen 13 II-4, a kill-butchery site dating from the northwest European Lower Palaeolithic. Journal of Taphonomy. 2008;6(2):71–141. Available from: http://doi.wiley.com/10.1002/9781118659991

25. Voormolen B. Ancient Hunters, Modern Butchers [PhD Thesis]. 2008. 

30. Serangeli J, Conard NJ. The behavioral and cultural stratigraphic contexts of the lithic assemblages from Schöningen. Journal of Human Evolution. 2015 Dec;89:287–97.

42. Serangeli J, Rodríguez-Álvarez B, Tucci M, Verheijen I, Bigga G, Böhner U, et al. The Project Schöningen from an ecological and cultural perspective. Quaternary Science Reviews [Internet]. 2018 Oct;198:140–55.

44. Sandgathe DM, Hayden B. Did Neanderthals eat inner bark? Antiquity [Internet]. 2003;77(298):709–18. Available from: https://www.cambridge.org/core/product/identifier/S0003598X00061652/type/journal_article

63. Martínez-Sancho E, Slámová L, Morganti S, Grefen C, Carvalho B, Dauphin B, et al. The GenTree Dendroecological Collection, tree-ring and wood density data from seven tree species across Europe. Sci Data [Internet]. 2020 Jan 2 [cited 2023 May 11];7(1):1. Available from: https://www.nature.com/articles/s41597-019-0340-y

64. Keunecke D, Evans R, Niemz P. Microstructural Properties of Common Yew and Norway Spruce Determined with Silviscan. IAWA J [Internet]. 2009 [cited 2023 May 9];30(2):165–78. Available from: https://brill.com/view/journals/iawa/30/2/article-p165_6.xml

65. Ravoajanahary T, Mothe F, Longuetaud F. A method for estimating tree ring density by coupling CT scanning and ring width measurements: application to the analysis of the ring width–ring density relationship in Picea abies trees. Trees [Internet]. 2023 Jun 1 [cited 2023 May 12];37(3):653–70. Available from: https://doi.org/10.1007/s00468-022-02373-2

73. Milks A. Yew wood, would you? An exploration of the selection of wood for Pleistocene spears. In: Berihuete-Azorín M, Seijo MM, López-Bultó O, Piqué R, editors. The missing woodland resources: Archaeobotanical studies of the use of plant raw materials [Internet]. Barkhuis; 2022 [cited 2023 Apr 26]. p. 1–18. Available from: http://www.jstor.org/stable/10.2307/j.ctv23wf366

77. Bordes L. X-Ray Tomography and Infrared Spectrometry for the Analysis Of Throwing Sticks & Boomerangs. Journal EXARC. 2019 Jun 18;3. 

92. Hrnčíř V. The Use of Wooden Clubs and Throwing Sticks among Recent Foragers: Cross-Cultural Survey and Implications for Research on Prehistoric Weaponry. Hum Nat [Internet]. 2023 Mar [cited 2023 May 5];34(1):122–52. Available from: https://link.springer.com/10.1007/s12110-023-09445-3

93. Davidson DS. Australian Throwing-Sticks, Throwing-clubs, and boomerangs. American Anthropologist. 1936;38(1):76–100. 

104. Gaudzinski-Windheuser S, Noack ES, Pop E, Herbst C, Pfleging J, Buchli J, et al. Evidence for close-range hunting by last interglacial Neanderthals. Nature Ecology & Evolution [Internet]. 2018 Jun;1087–92. Available from: http://dx.doi.org/10.1038/s41559-018-0596-1

105. Shea J, Sisk M. Complex projectile technology and Homo sapiens dispersal into western Eurasia. PaleoAnthropology. 2010;2010:100–22. 

109. Devereux G. La chasse collective au lapin chez les Hopi, Oraibi, Arizona. jsa [Internet]. 1941 [cited 2023 May 17];33(1):63–90. Available from: https://www.persee.fr/doc/jsa_0037-9174_1941_num_33_1_3796

114. Sahle Y, Ahmed S, Dira SJ. Javelin use among Ethiopia’s last indigenous hunters: Variability and further constraintson tip cross-sectional geometry. Journal of Anthropological Archaeology [Internet]. 2023 Jun 1 [cited 2023 Apr 13];70:101505. Available from: https://www.sciencedirect.com/science/article/pii/S0278416523000211

115. Spencer WB. Native tribes of the Northern territory of Australia. London: Macmillan and Co.; 1914. 

116. Morris J. Relationship between the British and the Tiwi in the vicinity of Port Dundas, Melville Island. Historical Society of the Northern Territory. 1964; 

117. Lloyd GT. Thirty-three Years in Tasmania and Victoria: Being the Actual Experience of the Author, Interspersed with Historic Jottings, Narratives, and Counsel to Emigrants . London: Houlsten and Wright; 1862. 

118. Roth HL. The aborigines of Tasmania. London: Kegan Paul, Trench, Trübner & Co; 1890. 

119. Robinson GA. Friendly mission : the Tasmanian journals and papers of George Augustus Robinson 1829-1834. Plomley NJB, editor. Hobart: Tasmanian Historical Society Research Association; 1966. 

120. Giles E. Australia Twice Traversed. Adelaide: Libraries Board of South Australia; 1889. 

121. Christison R, Edge-Partington J. 19. Notes on the Weapons of the Dalleburra Tribe, Queensland, Lately Presented to the British Museum by Mr. Robert Christison. Man [Internet]. 1903;3:37. Available from: http://www.jstor.org/stable/2839962?origin=crossref

122. Baker SW. Ismailia. Vol. 1. London: Macmillan; 1874.

---

## [Editor Report · Decision Letter 1]

12 Jun 2023

A double-pointed wooden throwing stick from Schöningen, Germany:  results and new insights from a multianalytical study

PONE-D-22-35734R1

Dear Dr. Milks,

We’re pleased to inform you that your revised manuscript has been judged scientifically suitable for publication and will be formally accepted for publication once it meets all outstanding technical requirements.

Kind regards,

Marco Peresani

Academic Editor

PLOS ONE
---

## [Editor Report · Acceptance letter]

22 Jun 2023

PONE-D-22-35734R1 

A double-pointed wooden throwing stick from Schöningen, Germany:  results and new insights from a multianalytical study 

Dear Dr. Milks:

I'm pleased to inform you that your manuscript has been deemed suitable for publication in PLOS ONE. Congratulations! Your manuscript is now with our production department. 

Kind regards, 

on behalf of

Dr. Marco Peresani 

Academic Editor

PLOS ONE